# Independence Test for Linear Non-Gaussian Data and Applications in Causal Discovery

**Yiqing Li**[1]    **Xiaofei Wang**[1,2]    **Boyang Sun**[1]    **Yewei Xia**[3]    **Kun Zhang**[1,4]
[1]Mohamed bin Zayed University of Artificial Intelligence    [2]Northeast Normal University
[3]Fudan University          [4]Carnegie Mellon University

## Abstract

Independence testing involves determining whether two variables are independent based on observed samples, which is a fundamental problem in causal discovery. Existing testing methods, such as HSIC, can theoretically detect broad forms of dependence, but may sacrifice statistical power when applied to limited samples with background knowledge of the distribution. In this paper, we focus on the linear non-Gaussian data, a widely supported model in scientific data analysis and causal discovery, where variables are linked linearly with noise terms that are non-Gaussian distributed. We provide a new theoretical characterization of independence in this case, showing that constancy of the conditional mean and variance is sufficient to guarantee independence under linear non-Gaussian models. Building on this result, we develop a kernel-based testing framework with provable asymptotic guarantees. Extensive experiments on synthetic and real-world datasets demonstrate that our method achieves higher power than existing approaches and significantly improves downstream causal discovery performance.

## 1 Introduction

Testing for statistical independence, i.e., deciding whether two variables are independent from observed samples, is fundamental in machine learning applications, such as in self-supervised representation learning (Li et al., 2021), feature selection (Candes et al., 2018), and in causal discovery (Spirtes et al., 2000). Over the decades, a rich toolbox of independence tests has emerged targeting different scenarios. Classical parametric methods include Pearson's correlation, Spearman's rank correlation, and Kendall's tau, etc. Recent advance focuses on nonparametric methods like HSIC (Gretton et al., 2005a) and dCor (Székely et al., 2007).In this paper, we aim to test the independence between linear mixtures of independent non-Gaussian components. This test is rather important for causal discovery in linear non-Gaussian models both with and without latent variables.

The assumption of a linear model is prevalent in causal discovery algorithms. Without structural assumptions on the data generation process, the causal direction is not identifiable from observational data (Spirtes et al., 2000; Pearl, 2009). Following a linear non-Gaussian acyclic model (LiNGAM), the Direct-LiNGAM algorithm (Shimizu et al., 2011) suggests an independence test scheme on regressor and residual to determine causal directions under the no latent confounders assumption. Furthermore, even when latent confounders are present, the causal direction can still be identified through independence testing. Specifically, the recently proposed generalized independence noise (GIN) (Xie et al., 2020; 2022) condition provides an elegant and efficient way to identify the existence of latent variables and recover the causal orders of the latent variables in LiNGAM model. Verifying the GIN condition involves a significant number of independence tests, strengthening the need for a reliable independence test in linear non-Gaussian settings.

Given the proven success of linear non-Gaussian models in solving real-world problems across various domains (Dong et al., 2023), surprisingly, to our knowledge there is no independence test specifically designed for the linear, non-Gaussian regime. In practice, researchers often resort to general-purpose non-parametric independence tests (e.g., HSIC). While these methods control Type I error, their generality can be a liability in our specific context. They are designed for arbitrary dependencies and may lack statistical power for the structured relationships generated by linear non-Gaussian models. This creates a methodological mismatch: while we employ identifiable models

that leverage non-Gaussianity, we rely on tests that do not exploit this structure—akin to using a shotgun to shoot a butterfly, which is inefficient and potentially ineffective.

A natural and pressing question arises: How can we design an independence test that is tailored to this well-established model class? By incorporating the model assumptions directly into the testing procedure, we can develop a method that is not only statistically more efficient but also conceptually simpler. This paper addresses this exact need. We propose a novel independence test specifically designed for the linear non-Gaussian data. We begin by providing a new characterization of independence in this setting. We show that, interestingly, for judging the independence of linear non-Gaussian data, it is enough to check the constancy of the conditional mean and the conditional variance. Based on this new characterization, we further designed a statistic that can test the conditions simultaneously. With derived asymptotic distributions, our method leverages the model constraints to achieve higher statistical power than generic alternatives, thereby providing a more robust foundation for causal discovery algorithms.

We summarize our contributions as follows.

- We propose a novel characterization of independence for linear mixtures of independent non-Gaussian components using only the conditional mean and conditional variance.

- We propose a statistic and derive its corresponding asymptotic distributions to test independence. We also prove the equivalence of the statistic and the independence characterization.

- We conduct extensive experiments on both synthetic and real-world data, which demonstrate the efficacy of our method. In addition, we integrate our testing method into an existing causal discovery algorithm and it outperforms other testing methods.

## 2 BACKGROUND

**Problem Definition.** For random variables $X \in \mathcal{X}$ and $Y \in \mathcal{Y}$, where $\mathcal{X}$ and $\mathcal{Y}$ are their domains, we say $X$ and $Y$ are independent if $\mathbb{P}_{XY} = \mathbb{P}_X \mathbb{P}_Y$, denoted by $X \perp\!\!\!\perp Y$. Given a dataset $\mathcal{D} = \{(x_i, y_i)\}_{i=1}^n$, where the pairs are independently and identically sampled from the joint distribution $\mathbb{P}_{XY}$, an independence test constructs a test statistic $T$ based on $\mathcal{D}$ to test for hypotheses:

$$\mathcal{H}_0 : X \perp\!\!\!\perp Y \qquad \text{v.s.} \qquad \mathcal{H}_1 : X \not\perp\!\!\!\perp Y.$$

The statistic $T$ is then compared with a critical value to decide whether to reject the null hypothesis $\mathcal{H}_0$. The quality of an independence test is typically characterized by two quantities: the probability of incorrectly rejecting $\mathcal{H}_0$ when it is true (Type I error), and the probability of failing to reject $\mathcal{H}_0$ when it is false (Type II error). An ideal test maintains the Type I error at a user-specified significance level $\alpha$, while achieving high statistical power ($1-$ Type II error rate).

A direct way to check for independence based on the definition. That is, estimate the probability densities of the joint distribution $\mathbb{P}_{XY}$ and the marginal distribution $\mathbb{P}_X, \mathbb{P}_Y$, and then evaluate if $\mathbb{P}_{XY} = \mathbb{P}_X \mathbb{P}_Y$ is satisfied almost surely. For example, mutual information measures the dependence strength between two variables using the KL divergence between $\mathbb{P}_{XY}$ and $\mathbb{P}_X \mathbb{P}_Y$. However, estimating the probability densities from finite samples is difficult. Some distributions may even have no densities, which may further deteriorate the testing performance. Instead, (Jacod & Protter, 2004) provides an alternative characterization of the independence of random variables.

**Lemma 2.1.** *The random variables $X$ and $Y$ are independent if and only if $\mathbb{C}ov(f(X), g(Y)) = 0$ for each pair $(f, g)$ of bounded, continuous functions, i.e. $f \in \mathcal{C}_b(\mathcal{X})$ and $g \in \mathcal{C}_b(\mathcal{Y})$.*

Lemma 2.1 provides a direct test criterion without the need for an intermediate density estimator. However, the space of bounded, continuous functions is too rich, which will raise the consistency issue. That is, the empirical estimate converges slowly to its expectation as the sample size increases. Instead, To address this, one may restrict attention to a more manageable yet expressive function class. In particular, kernel-based methods provide a principled way to address this by restricting to an RKHS, which is both manageable and still rich enough to capture independence.

**Reproducing Kernel Hilbert Space.** A kernel function $k(x, x')$ is defined as a symmetric, positive definite mapping $k : \mathcal{X} \times \mathcal{X} \to \mathbb{R}$, which admits a representation in terms of an inner product,

$k(x, x') = \langle \phi(x), \phi(x') \rangle_{\mathcal{H}}$, where $\phi(x)$ is a feature map in a Hilbert space $\mathcal{H}$. Furthermore, we say that $\mathcal{H}$ is a reproducing kernel Hilbert space (RKHS) if $\mathcal{H}$ is a Hilbert space of functions $f : \mathcal{X} \to \mathbb{R}$ that satisfies the reproducing property $\langle \phi(x), f \rangle_{\mathcal{H}} = f(x), \forall f \in \mathcal{H}$. A linear operator $A : \mathcal{G} \to \mathcal{F}$, where $\mathcal{G}, \mathcal{F}$ are separable Hilbert spaces (i.e. the Hilbert space has a countable orthonormal basis), is called a Hilbert-Schmidt operator if it has a finite Hilbert-Schmidt norm, $\|A\|_{\mathcal{HS}}^2 = \sum_{j \in J} \|Ag_j\|_{\mathcal{F}}^2$, where $\{g_j\}_{j \in J}$ denotes any orthonormal basis of $\mathcal{G}$. In finite-dimensional Euclidean spaces with the linear kernel, this reduces to the Frobenius norm of a matrix. See this link for more details.

**Definition 2.2** (Universal Kernel). A continuous kernel $k(\cdot, \cdot)$ on a compact metric space $(\mathcal{X}, d)$ is called universal if and only if the RKHS $\mathcal{F}$ induced by the kernel is dense in $\mathcal{C}_b(\mathcal{X})$, i.e. for every function $f \in \mathcal{C}_b(\mathcal{X})$ and every $\epsilon > 0$ there exists a function $g$ induced by $k$ with $\|f - g\|_\infty \leq \epsilon$.

Universal kernel provides a smaller space than $\mathcal{C}_b(\mathcal{X})$ to consider the functions while keeping the characterization property, as used in COCO (Gretton et al., 2005b) and HSIC (Gretton et al., 2005a). These methods connect independence with the zero-value of their statistics, when universal kernels are used on the compact domains $\mathcal{X}$ and $\mathcal{Y}$ (or more generally, characteristic kernels[1]).

**Definition 2.3.** (Gretton et al., 2007) The Hilbert-Schmidt Independence Criterion between $X$ and $Y$, denoted as $\mathrm{HSIC}(X, Y)$, is the HS norm of the covariance operator

$$\|\Sigma_{XY}\|_{\mathcal{HS}}^2 = \|\mathbb{E}_{\mathbb{P}_{XY}}[(\psi_X - \mu_X) \otimes (\phi_Y - \mu_Y)]\|_{\mathcal{HS}}^2,$$

where $\mu_X \triangleq \mathbb{E}_{\mathbb{P}_X}[\psi(X)]$, $\mu_Y \triangleq \mathbb{E}_{\mathbb{P}_Y}[\phi(Y)]$, and $\otimes$ is the tensor product.

When using characteristic kernels, $X \perp\!\!\!\perp Y$ if and only if $\mathrm{HSIC}(X, Y) = 0$.

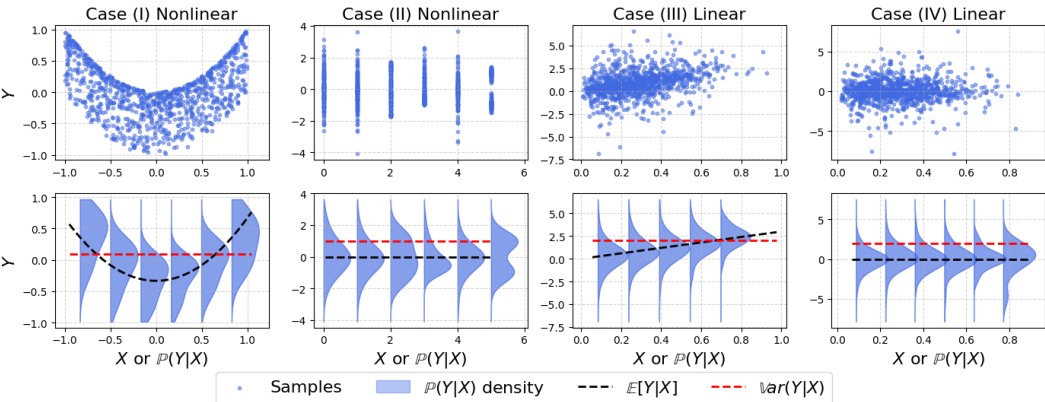

Figure 1: Illustration of the motivation. The first row shows the **scatter plots** between $X$ and $Y$, and the second row gives plots the **conditional densities**, $\mathbb{P}(Y \mid X)$, for each bin (continuous) or value (discrete) of $X$. For clear comparison, the conditional mean $\mathbb{E}(Y \mid X)$ and the conditional variance $\mathbb{V}ar(Y \mid X)$ are drawn in red and black dashed lines respectively. Data generation process: (I) $X = U, Y = U^2 - V^2$, where $U, V \sim \mathcal{U}(-1, 1)$; (II) $X \sim \mathcal{U}(\{1, ..., 6\})$, with each value of $X$ $Y$ follows $\mathcal{N}(0, 1), \mathrm{Laplace}(0, \frac{1}{\sqrt{2}}), \mathcal{U}(-\sqrt{3}, \sqrt{3}), \mathrm{Exp}(1) - 1, t_{\nu=3}$, and $0.5 \cdot \mathcal{N}(-2, 0.25) + 0.5 \cdot \mathcal{N}(2, 0.25)$ respectively, and $Y$ is normalized so that $\mathbb{V}ar(Y \mid X) \equiv 1$; (III) $Y = 3 \cdot X + \epsilon$ and $X, \epsilon \sim \mathrm{Beta}(2, 5)$ independently; (IV) $X \sim \mathrm{Beta}(2, 5)$ and $Y \sim \mathrm{Laplace}(0, 1)$. Note that for both Case (II) and (IV) we have $\mathbb{E}(Y \mid X)$ and $\mathbb{V}ar(Y \mid X)$ are controlled as constants.

## 3 MOTIVATION

Recall that classical independence tests designed for Gaussian data, such as Fisher's $z$ test, rely only on the first two moments of the variables. With the assumption of a simple linear model $Y = \beta X + \alpha$, independnece can be assessed by testing the null hypothesis $\beta = 0$, where $\beta = \frac{\mathbb{C}ov(X, Y)}{\mathbb{V}ar(X)}$,

---

[1]We give the formal definition of characteristic kernels in the appendix.

which again only depends on the first- and second-order moments. These are possible because for Gaussian distributions, dependence is fully characterized by the mean and covariance structure; for the simple linear model, the relationship is also constrained to these low-order moments. Motivated by this, a natural question arises in the context of linear mixtures of non-Gaussian sources: can independence between $X$ and $Y$ also be determined from low-order moment information, e.g. the first and second conditional moments?

Figure 1 presents a preliminary exploration and illustration. Case (I) and Case (II) are nonlinear relationships while Case (III) and Case (IV) contain linear relations only. We constrain the conditional mean $\mathbb{E}(Y \mid X)$ and conditional variance $\mathbb{V}ar(Y \mid X)$ to be constant in Case (II) and (IV), and leave them free in Case (I) and (III). For nonlinear data, there exist situations in which $X$ and $Y$ are still dependent after enforcing the constancy of the conditional mean and variance. For example, in Case (II) we can clearly see the skewness and kurtosis of $\mathbb{P}(Y \mid X)$ change across different values of $X$, manifesting the dependence between $X$ and $Y$ given the conditions. However, if $X$ and $Y$ are linear mixtures of independent non-Gaussian variables, as shown in Case (IV), it is hard, or even not possible to keep $X$ and $Y$ dependent while keeping $\mathbb{E}(Y \mid X)$ and $\mathbb{V}ar(Y \mid X)$ constants.

Imposing a tighter model class typically affords higher power. In the linear non-Gaussian setting, the observed constancy of the first two conditional moments provides a precise handle for testing. Building on this, we develop an independence test specialized to linear mixtures of independent non-Gaussian components that outperforms generic nonparametric tests when the assumptions hold, leading to more accurate estimation of linear non-Gaussian causal graphs.

## 4 METHOD

Based on the surprising observation in Section 3, we now formally formulate an independence test that utilizes the information in the model class. Specifically, we first give a new characterization of independence for linear mixtures of independent non-Gaussian components, as shown in Theorem 4.2. Moreover, we build the connection between the conditions in the new characterization and the uncorrelatedness of the first and second order information of $Y$, which is shown in Theorem 4.3, and give the corresponding statistic. Finally, we also derive the asymptotic distributions of the statistic and the estimation of the testing threshold under $\mathcal{H}_0$ in Theorem 4.5, 4.6, and 4.7.

### 4.1 CHARACTERIZATION OF INDEPENDENCE FOR LINEAR MIXTURES

We provide what appears to be the first explicit characterization of independence for linear mixtures of independent non-Gaussian components. We first introduce a lemma which would be useful later:

**Lemma 4.1.** *The following two statements for the random vector $(X, Y)$ are equivalent:*

$$(i) \qquad \mathbb{E}(Y \mid X) = \alpha + \beta X, \quad \mathbb{V}ar(Y \mid X) = \sigma^2 = \text{constant},$$

$$(ii) \quad \begin{cases} \frac{\partial \phi(t_1, t_2)}{\partial t_2}\Big|_{t_2=0} = i\alpha\phi(t_1, 0) + \beta\frac{d\phi(t_1,0)}{dt_1} \\ \frac{\partial^2 \phi(t_1, t_2)}{\partial t_2^2}\Big|_{t_2=0} = -\left(\sigma^2 + \alpha^2\right)\phi(t_1, 0) + 2i\alpha\beta\frac{d\phi(t_1,0)}{dt_1} + \beta^2\frac{d^2\phi(t_1,0)}{dt_1^2} \end{cases} . \qquad (1)$$

Here $\phi(t_1, t_2) = \mathbb{E}\left[e^{i(t_1 X + t_2 Y)}\right]$ is the joint characteristic function (c.f.) of $(X, Y)$. The proof of it is in appendix. This lemma connects the regression conditional with simple analytic identities of the joint characteristic function. This technical tool allows us to derive the following main theorem:

**Theorem 4.2.** *Let $\varepsilon_1, \ldots, \varepsilon_m$ be independent, non-Gaussian random variables with finite variances, and let $Y = \sum_{j=1}^m a_j\varepsilon_j, X = \sum_{j=1}^m b_j\varepsilon_j$ be two linear mixtures of $\varepsilon_1, \ldots, \varepsilon_m$ with coefficients $\{a_j\}_{j=1}^m$ and $\{b_j\}_{j=1}^m$, respectively. $\forall j \in [m]$, $a_j^2 + b_j^2 > 0$. Then $Y \perp\!\!\!\perp X$ if and only if there exist constants $c \in \mathbb{R}$ and $\sigma_0^2 \geq 0$ such that*

(i) *(**Constancy of regression**)* $\mathbb{E}(Y \mid X) = c$,

(ii) *(**Homoscedasticity**)* $\mathbb{V}ar(Y \mid X) = \mathbb{E}[Y^2 \mid X] - (\mathbb{E}[Y \mid X])^2 = \sigma_0^2$.

*Proof sketch.* The necessity is immediate. For sufficiency, assume (i)–(ii). First decompose

$$Y = L + A, \quad L = \sum_{i \in \mathcal{S}} a_i \varepsilon_i, \ A = \sum_{i \notin \mathcal{S}} a_i \varepsilon_i; \qquad X = M + B, \quad M = \sum_{i \in \mathcal{S}} b_i \varepsilon_i, \ B = \sum_{i \notin \mathcal{S}} b_i \varepsilon_i,$$

where $\mathcal{S} = \{i : a_i b_i \neq 0\}$ are the index of the common components shared by $X$ and $Y$, and $A$ and $B$ are linear mixtures of the components specific to $Y$ and $X$. Suppose $\mathcal{S} \neq \emptyset$. Since the $\varepsilon_i$ are mutually independent, $A, B$ are independent and also independent of $L, M$, according to Darmois-Skitovich Theorem. Given (i)–(ii) and $A \perp\!\!\!\perp X$ we obtain constants $\alpha, \sigma^2$ such that

$$\mathbb{E}(L \mid X) = \mathbb{E}(L \mid M) = \alpha, \qquad \mathbb{V}ar(L \mid X) = \mathbb{V}ar(L \mid M) = \sigma^2.$$

Let $f_j$ denote the c.f. of $\varepsilon_j$ and $\theta_j = \log f_j$ near the origin. Let $\phi(t_1, t_2) = \mathbb{E}\left[e^{i(t_1 M + t_2 L)}\right]$ be the joint c.f. of $(M, L)$. Applying Lemma 4.1 with $\beta = 0$ and evaluating at $t_2 = 0$ yields, for all $t \in \mathbb{R}$, $\sum_{j \in \mathcal{S}} a_j \theta_j'(b_j t) = i\alpha$, $\sum_{j \in \mathcal{S}} a_j^2 \theta_j''(b_j t) = -\sigma^2$. Integrating the second identity twice and using $\theta_j(0) = 0$ and $\theta_j'(0) = i\mathbb{E}[\varepsilon_j]$ gives $\sum_{j \in \mathcal{S}} \frac{a_j^2}{b_j^2} \theta_j(b_j t) = i\mu t - \frac{1}{2}\sigma^2 t^2$ for some constant $\mu$. Exponentiating both sides, $\prod_{j \in \mathcal{S}} \left[f_j(b_j t)\right]^{a_j^2/b_j^2} = \exp\left(i\mu t - \frac{1}{2}\sigma^2 t^2\right)$, whose right-hand side is a Gaussian c.f. with the Hermitian property. By the $\alpha$-decomposition Theorem, each $f_j$ ($j \in \mathcal{S}$) must itself be a Gaussian c.f., hence the corresponding $\varepsilon_j$ are Gaussian. This contradicts the non-Gaussian assumption unless $\mathcal{S} = \varnothing$. Consequently $X$ and $Y$ share no common source, so $X \perp\!\!\!\perp Y$. $\qquad\square$

This theorem guaranties that, in the linear non-Gaussian setting, independence can be decided by examining only the first two conditional moments, $\mathbb{E}(Y \mid X)$ and $\mathbb{V}ar(Y \mid X)$, even though outside this setting higher-order information is often needed since only Gaussian distributions have all cumulants[2] of order $\geq 3$ equals to zero. Recall the comparison in Figure 1, these two conditions alone do not exclude more complex forms of dependence in general nonlinear non-Gaussian models, showing the importance of linearity.

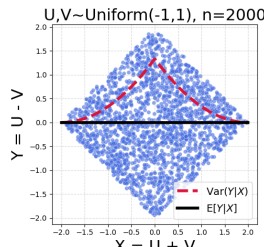

We also want to emphasize that under the linear non-Gaussian model our criterion is jointly necessary and sufficient. Although $X$ and $Y$ are linear mixtures, checking only the conditional mean is insufficient, which might be counterintuitive. Neither $\mathbb{E}(Y \mid X)$ being constant nor $\mathbb{V}ar(Y \mid X)$ being constant alone implies independence. Figure 2 shows an example. We can easily construct two dependent variables $(X, Y)$, $X = U + V$, $Y = U - V$, where $U$ and $V$ are two independent non-Gaussian variables with zero mean (here $U, V \sim \mathcal{U}(-1, 1)$). $\mathbb{E}(Y \mid X)$ is a constant, while $\mathbb{V}ar(Y \mid X)$ varies. This also explains why a single linear regression of $Y$ on $X$ cannot serve as a valid independence test for linear

Figure 2: An example that the constancy of regression holds while homoscedasticity does not. The conditional mean $\mathbb{E}(Y \mid X)$ and variance $\mathbb{V}ar(Y \mid X)$ are the black and red lines in the figure, respectively. Clearly $X \not\perp\!\!\!\perp Y$.

non-Gaussian data. $\mathcal{H}_0 : \beta = 0$ only considers the conditional mean and along linear alternatives, i.e., $\beta \neq 0$. Dependence can manifest either in the conditional variance or in nonlinear mean effects.

## 4.2 DERIVATION OF THE STATISTICS

Based on this characterization, we reduce the problem of testing independence to testing whether the conditional mean and conditional variance are constant. One may view $\mathbb{E}(Y \mid X)$ and $\mathbb{V}ar(Y \mid X)$ as functions of $X$, and there exist nonparametric procedures for testing whether such functions are constant, e.g., (Bierens, 1990; Fan & Jiang, 2007). However, these methods are computationally demanding and, moreover, require separate tests for both $Y$ and $Y^2$ to verify the two conditions in Theorem 4.2. Interestingly, we show that the two conditions can in fact be tested simultaneously with a single statistic. To this end, we introduce an intermediate representation of independence for linear mixtures of independent non-Gaussian components. The proof is deferred to Appendix C.4.

**Theorem 4.3.** *Let $\varepsilon_1, \ldots, \varepsilon_m$ be independent, non-Gaussian random variables with $\mathbb{E}[\varepsilon_i^2] < \infty$ for all $i$. Define $Y = \sum_{i=1}^m a_i \varepsilon_i$ and $X = \sum_{i=1}^m b_i \varepsilon_i$. Then $X \perp\!\!\!\perp Y$ if and only if for any bounded, continuous function $f$, $\mathbb{C}ov(f(X), Y) = 0$ and $\mathbb{C}ov(f(X), Y^2) = 0$.*

---

[2]Cumulants are defined as $\kappa_r(X) := \frac{1}{i^r} \frac{d^r}{dt^r} \log \phi_X(t)\big|_{t=0}$ whenever the derivative exists. They are polynomial combinations of moments (e.g. $\kappa_1 = \mathbb{E}[X]$, $\kappa_2 = \text{Var}(X)$). $\kappa_r = 0$ for all $r \geq 3$ iff $X$ is Gaussian.

*Remark* 4.4. The implication of the sufficiency relies on the linear non-Gaussian structure above; without it, constant conditional mean and variance do not imply independence in general.

Based on Theorem 4.3, it suffices to check that, for every bounded continuous function $f$, $\text{Cov}(f(X), Y) = 0$ and $\text{Cov}(f(X), Y^2) = 0$. For implementation, we adopt a kernel criterion. Let $k$ be a universal kernel on $X$ (e.g. Gaussian), and let $l$ be the degree-2 polynomial kernel on $Y$ with RKHS $\mathcal{H}_l$ involving the functions $y \mapsto y$ and $y \mapsto y^2$. $\phi, \psi$ are their feature maps. We define the first variant of Linear Non-Gaussian Independence Criterion (LiNGIC$_1$) as

$$\text{LiNGIC}_1(X, Y) = \|\mathbb{C}ov(\phi(X), \psi(Y))\|_{\mathcal{HS}}^2 = \left\|\mathbb{E}[(\phi(X) - \mu_X^k) \otimes (\psi(Y) - \mu_Y^l)]\right\|_{\mathcal{HS}}^2,$$

where $\mu_X^k \triangleq \mathbb{E}[\phi(X)]$, $\mu_Y^l \triangleq \mathbb{E}[\psi(Y)]$, and $\otimes$ is the tensor product. Note this construction coincides with HSIC equiped with gaussian and polynomial kernels, though polynomial kernels are not characteristic thus seldom used in the literature. In our case, degree-2 polynomial kernel on one side suffices and $\text{LiNGIC}(X, Y) = 0$ if and only if $\forall f \in \mathcal{C}_b(X)$, $\text{Cov}(f(X), Y) = 0$ and $\text{Cov}(f(X), Y^2) = 0$, which is equivalent to $X \perp\!\!\!\perp Y$ for linear non-Gaussian data by Theorem 4.3.

Given a dataset $\mathcal{D} = \{(x_i, y_i)\}_{i=1}^n$, a biased estimator of LiNGIC$_1(X, Y)$ is[3]

$$\text{LiNGIC}_{1b}(\mathcal{D}) = \frac{1}{n^2} \sum_{i,j}^n k_{ij}^X l_{ij}^Y + \frac{1}{n^4} \sum_{i,j,q,r}^n k_{ij}^X l_{qr}^Y - 2\frac{1}{n^3} \sum_{i,j,q}^n k_{ij}^X l_{iq}^Y = \frac{1}{n^2} \text{Tr}(\boldsymbol{K}_X \boldsymbol{H} \boldsymbol{L}_Y \boldsymbol{H}),$$

where $k_{ij}^X := \sum_{i,j} k(x_i, x_j)$, $l_{ij}^Y := \sum_{i,j} l(y_i, y_j)$, $\boldsymbol{K}_X$ is the $n \times n$ matrix with entries $k_{ij}^X$ and $\boldsymbol{L}_Y$ with entries $l_{ij}^Y$, $\boldsymbol{H} = \mathbf{I} - \frac{1}{n}\mathbf{1}\mathbf{1}^\top$, and $\mathbf{1}$ is a $n \times 1$ vector of ones. This empirical statistic can be derived directly using plug-in estimation method and is a sum of three V-statistics (Serfling, 1980).

However, this criterion is asymmetric w.r.t. $(X, Y)$. Since polynomial kernels are unbounded on non-compact domains, in practice this can lead to numerical instability when $Y$ takes extreme values, especially under heavy-tailed distributions. See Appendix E for empirical observation. To symmetrize our statistic, we further design the following feature maps. If we write the feature map in matrix form $\boldsymbol{\varphi}^1(\cdot)$, we define the extended feature map $\varphi^1$ and its kernel matrix $\mathring{\boldsymbol{K}}^1$ as

$$\boldsymbol{\varphi}^1(\boldsymbol{x}) = \begin{bmatrix} \boldsymbol{\phi}(\boldsymbol{x}) & \mathbf{0} \\ \mathbf{0} & \boldsymbol{\psi}(\boldsymbol{x}) \end{bmatrix}, \quad \mathring{\boldsymbol{K}}_X^1 = \boldsymbol{\varphi}^1(\boldsymbol{x})\boldsymbol{\varphi}^1(\boldsymbol{x})^T = \begin{bmatrix} \boldsymbol{K}_X & \mathbf{0} \\ \mathbf{0} & \boldsymbol{L}_X \end{bmatrix},$$

where we use $\boldsymbol{x} = [x_1, ... x_n]^T$ and $\boldsymbol{y} = [y_1, ... y_n]^T$ to represent vectors of samples. $\varphi^2$ and $\mathring{\boldsymbol{K}}^2$ are defined similarly with an exchange of place of $\phi$ and $\psi$. It can be easily verified that $\left\|\mathbb{C}ov(\varphi^1(X), \varphi^2(Y))\right\|_{\mathcal{HS}}^2 = \left\|\mathbb{C}ov(\phi(X), \psi(Y))\right\|_{\mathcal{HS}}^2 + \left\|\mathbb{C}ov(\psi(X), \phi(Y))\right\|_{\mathcal{HS}}^2$. That is, using these feature maps is equivalent to combining two directions, LiNGIC$_1(X, Y)$ and LiNGIC$_1(Y, X)$, directly together. Lastly, we use $\text{LiNGIC}(X, Y) \triangleq \left\|\mathbb{C}ov(\varphi^1(X), \varphi^2(Y))\right\|_{\mathcal{HS}}^2$ as our final criterion, which is symmetric to $(X, Y)$. And now the biased estimation of the statistic becomes $\text{LiNGIC}_b(\mathcal{D}) = \frac{1}{n^2} \text{Tr}(\boldsymbol{K}_X \boldsymbol{H} \boldsymbol{L}_Y \boldsymbol{H}) + \frac{1}{n^2} \text{Tr}(\boldsymbol{K}_Y \boldsymbol{H} \boldsymbol{L}_X \boldsymbol{H})$.

## 4.3 ASYMPTOTIC DISTRIBUTION AND ITS APPROXIMATION

We now describe the null distributions of the test statistic. Suppose $\mathcal{D} = \{w_i\}_{i=1}^n = \{(x_i, y_i)\}_{i=1}^n$. We first define a symmetric function that satisfies $\text{LiNGIC}_b(\mathcal{D}) = \frac{1}{n^4} \sum_{i,j,q,r}^n h_{ijqr}$ as

$$h_{ijqr} = \frac{1}{4!} \sum_{(t,u,v,w)}^{(i,j,q,r)} \{k_{tu}^X l_{tu}^Y + k_{tu}^X l_{vw}^Y - 2k_{tu}^X l_{tv}^Y + k_{tu}^Y l_{tu}^X + k_{tu}^Y l_{vw}^X - 2k_{tu}^Y l_{tv}^X\}, \tag{2}$$

where $k_{ij}^Y = k(y_i, y_j)$, $l_{ij}^X = l(x_i, x_j)$, the sum represents all ordered quadruples $(t, u, v, w)$ drawn without replacement from $(i, j, q, r)$, and assume $\mathbb{E}(h^2) < \infty$.

---

[3]We use a biased estimator instead of an unbiased for the purpose of computation efficiency. The unbiased version of the statistic can be easily obtained by replacing the V-statistics with U-statistics. As mentioned by (Gretton et al., 2005b), the biased version converges to the unbiased version at the rate $\mathcal{O}(n^{-1})$.

**Theorem 4.5** (Null distribution). *Under $\mathcal{H}_0$, we have $\mathbb{E}_i h_{ijqr} = 0$. In this case, $\text{LiNGIC}_b(\mathcal{D})$ converges in distribution to a weighted sum of $\mathcal{X}^2$ variables, i.e.,*

$$n \, \text{LiNGIC}_b(\mathcal{D}) \xrightarrow{d} \sum_{l=1}^{\infty} \lambda_l \chi_{1l}^2, \quad \lambda_l \ \text{satisfies} \ \lambda_l \psi_l(w_j) = \int h_{ijqr} \psi_l(w_i) dF_{i,q,r}. \tag{3}$$

*Here $\chi_{1l}^2$ are i.i.d. chi-square variables with freedom one. Since $w_i \triangleq (x_i, y_i)$, $\lambda_l$ are the solutions to the eigenvalue problem integrating over the distribution of variables $w_i, w_q$, and $w_r$.*

Next, we give a theorem about the asymptotic distribution when $\text{LiNGIC}_b(\mathcal{D}) > 0$, i.e., $X \not\perp\!\!\!\perp Y$. This distribution would be useful in analyzing consistency[4].

**Theorem 4.6.** *When $\text{LiNGIC}(X, Y) > 0$, $\text{LiNGIC}_b(\mathcal{D})$ converges in distribution to a Gaussian:*

$$\sqrt{n} \left( \text{LiNGIC}_b(\mathcal{D}) - \text{LiNGIC}(X, Y) \right) \xrightarrow{d} \mathcal{N}(0, \sigma^2). \tag{4}$$

*The variance $\sigma^2 = 16(\mathbb{E}_i(\mathbb{E}_{j,q,r} h_{ijqr})^2 - \text{LiNGIC}(X, Y)^2)$, where $\mathbb{E}_{j,q,r} \triangleq \mathbb{E}_{w_j, w_q, w_r}$.*

To use LiNGIC as a level-$\alpha$ hypothesis test, we need the $(1 - \alpha)$ critical value of its null distribution. The asymptotic null law in Equation (3) is an infinite weighted sum of $\chi^2$ variables and is not tractable to evaluate exactly. One may want to use the permutation test (Ernst, 2004), which permutes the ordering of the $Y$ sample and keeps $X$ fixed to ensure the independence between $X$ and $Y$ thus estimating the quantile. However, this can be computationally intensive for large $n$. As a faster alternative, one can approximate the null by a Gamma distribution (Kankainen, 1995), fitting shape and scale via moment matching as in (Gretton et al., 2005a; Zhang et al., 2012):

$$n \, \text{LiNGIC}_b(\mathcal{D}) \sim \text{Gamma}(\gamma, \beta), \ \text{where} \ \gamma = \frac{A^2}{B}, \ \beta = \frac{B}{A}.$$

Here $A \triangleq \mathbb{E}[n \cdot \text{LiNGIC}_b(\mathcal{D})]$ and $B \triangleq \mathbb{V}ar(n \cdot \text{LiNGIC}_b(\mathcal{D}))$. We estimate them as follow.

**Theorem 4.7.** *Under $\mathcal{H}_0$, the estimation of mean with bias of $\mathcal{O}(n^{-1})$ to $A$ can be given by*

$$\widehat{A} = \widehat{\mu_{xx}^k \mu_{yy}^l} + \widehat{\|\mu_x^k\|^2 \|\mu_y^l\|^2} - \widehat{\mu_{xx}^k \|\mu_y^l\|^2} - \widehat{\mu_{yy}^l \|\mu_x^k\|^2}$$
$$+ \widehat{\mu_{yy}^k \mu_{xx}^l} + \widehat{\|\mu_y^k\|^2 \|\mu_x^l\|^2} - \widehat{\mu_{yy}^k \|\mu_x^l\|^2} - \widehat{\mu_{xx}^l \|\mu_y^k\|^2},$$

*where $\mu_x^l \triangleq \mathbb{E}\psi(x)$, $\mu_y^k \triangleq \mathbb{E}\phi(y)$, $\mu_{xx}^k \triangleq \mathbb{E}k(x, x)$, $\mu_{xx}^l \triangleq \mathbb{E}l(x, x)$, $\mu_{yy}^k \triangleq \mathbb{E}k(y, y)$, and $\mu_{yy}^l \triangleq \mathbb{E}l(y, y)$. Also, the estimation of variance with bias of $\mathcal{O}(n^{-1})$ to $B$ can be given by*

$$\widehat{B} = \frac{2(n-4)(n-5)}{(n-1)^2(n-2)(n-3)} \sum_{i=1}^{3} \mathbf{1}^T (\mathbf{M}_i - \text{diag}(\mathbf{M}_i)) \mathbf{1},$$

*where $\mathbf{M}_i$ are ($\odot$ denotes the entry-wise matrix product and $M^{\cdot 2}$ the entry-wise matrix power)*

$$\mathbf{M}_1 = ((\boldsymbol{H}\boldsymbol{K}_X\boldsymbol{H}) \odot (\boldsymbol{H}\boldsymbol{L}_Y\boldsymbol{H}))^{\cdot 2}, \ \mathbf{M}_2 = ((\boldsymbol{H}\boldsymbol{K}_Y\boldsymbol{H}) \odot (\boldsymbol{H}\boldsymbol{L}_X\boldsymbol{H}))^{\cdot 2},$$
$$\mathbf{M}_3 = 2(\boldsymbol{H}\boldsymbol{K}_X\boldsymbol{H}) \odot (\boldsymbol{H}\boldsymbol{L}_X\boldsymbol{H}) \odot (\boldsymbol{H}\boldsymbol{K}_Y\boldsymbol{H}) \odot (\boldsymbol{H}\boldsymbol{L}_Y\boldsymbol{H}).$$

## 5 RELATED WORKS

Independence testing has long been an active research direction in statistics. Early approaches include the $F$-test (Tiku, 1967) and the Chi-squared test (Greenwood & Nikulin, 1996) for discrete or categorical variables. For continuous variables, the Pearson correlation coefficient (Benesty et al., 2009) is widely used, and in the linear Gaussian setting it fully characterizes independence. Another category involves nonparametric rank-based methods, which aim to achieve robustness against distribution shapes by focusing on the relative order of observations rather than their precise magnitudes, such as Spearman's rank correlation and Kendall's tau. More recently, (Chatterjee, 2021) proposed a rank-based correlation coefficient that equals zero if and only if two

---

[4] Whether the Type II error will converge to 0 as $n \to \infty$.

variables are independent and equals one only under functional dependence. (Han et al., 2017) developed distribution-free, rate-optimal tests for high-dimensional mutual independence based on rank statistics. Nevertheless, the reliance on rank-order statistics limits these approaches to detecting monotonic dependencies, failing to provide a general yet computationally efficient solution. Mutual information (Berrett & Samworth, 2019) provides a fully general characterization of independence, yet its practical use is constrained by the difficulty of accurate density estimation. To overcome these limitations, kernel-based independence tests (Bach & Jordan, 2002; Gretton et al., 2005b; 2003; 2005a) have been developed. A kernel function implicitly defines an inner product in a reproducing kernel Hilbert space (RKHS) (Berlinet & Thomas-Agnan, 2011), which induces a similarity measure between data points. A widely used example is HSIC (Gretton et al., 2005a), which quantifies dependence via the squared Hilbert-Schmidt norm of cross-covariance operators in RKHS. Variants such as random Fourier feature approximations (Zhang et al., 2018) improve computational efficiency, while sometimes at the cost of statistical power. Recent advances focus on improving previous methods (Ren et al., 2024), find tests that suit high-dimension data (Zhang & Zhu, 2024; Zhang et al., 2023) or time series data (Liu et al., 2023), address rare dependence patterns via adaptive sample reweighting (Li et al., 2025), etc. Besides, the random dependence coefficient (RDC) (Lopez-Paz et al., 2013) achieves marginal invariance through copula transformations and measures dependence by maximizing correlations under random projections, offering a computationally efficient solution. However, all these existing methods are not specifically designed for the linear non-Gaussian setting, which is crucial for many causal discovery frameworks.

## 6 EXPERIMENT

We test the proposed method on both synthetic and real data to compare its performance with other baselines. We also include the experiments that applying our method to causal discovery methods.

**Baselines.** All baselines follow their default settings unless otherwise stated. **HSIC** (Gretton et al., 2007): the original HSIC test using gamma approximation. **dCor** (Székely et al., 2007): A normalized covariance between the centered pairwise Euclidean distance matrices. **HSIC-RFF** (Zhang et al., 2018): HSIC using finite-dimensional random Fourier feature mappings to approximate kernels. **LFHSIC** (Ren et al., 2024): HSIC test with adaptively learned bandwidth.

For all methods that rely on a characteristic kernel, we adopt a Gaussian kernel whose bandwidth is selected heuristically according to the sample size, with LFHSIC being the only exception, as it learns its bandwidths directly from the data. Due to space limit, details about the experiment settings, experiments on more data generation processes, comparisons with more baseline methods, and real-world data experiments, please see Appendix E.

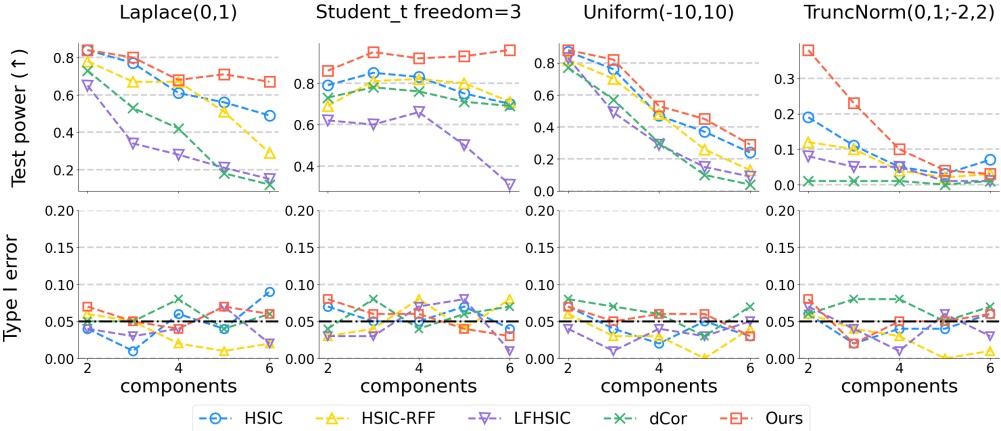

Figure 3: The experiment results when we change the number of the independent components of the linear mixtures with 500 samples. The number of components $d \in \{2, 3, 4, 5, 6\}$. Each column shows the results with $\varepsilon_i \sim$ a different distribution. The first row demonstrates Test Power and the second row shows the Type I error. The significance level $0.05$ is annotated as the black line.

## 6.1 SYNTHETIC DATA

**Data Generation.** We generate $n$ pairs of two linear mixtures, $Y = \sum_{i=1}^{m} a_i \varepsilon_i$ and $X = \sum_{i=1}^{m} b_i \varepsilon_i$, where $\varepsilon_i, i = 1, \ldots, m$ are independent and identically distributed non-Gaussian components. For simplicity, we restrict them as the same distribution, which is chosen from $\{$Laplace$(0,1)$, $t_{\nu=3}$, $\mathcal{U}(-10,10)$, TruncNorm$(0,1;-2,2)\}$. For power rate, the weights are randomly generated $a_i, b_i \sim \mathcal{U}(-1,1)$. We also ensure that the dependence between $X$ and $Y$ does not vanish due to randomness by constraining $a_i b_i \geq 0.1, \forall i$. Finally, we whitened the data to ensure zero correlation. For type I error, we make sure $X$ and $Y$ do not share any common components by $Y = \sum_{i=1}^{m} a_i \varepsilon_i$ and $X = \sum_{i=m+1}^{2m} b_i \varepsilon_i$, where $\varepsilon_i, i = 1, \ldots, 2m$, have the same distribution.

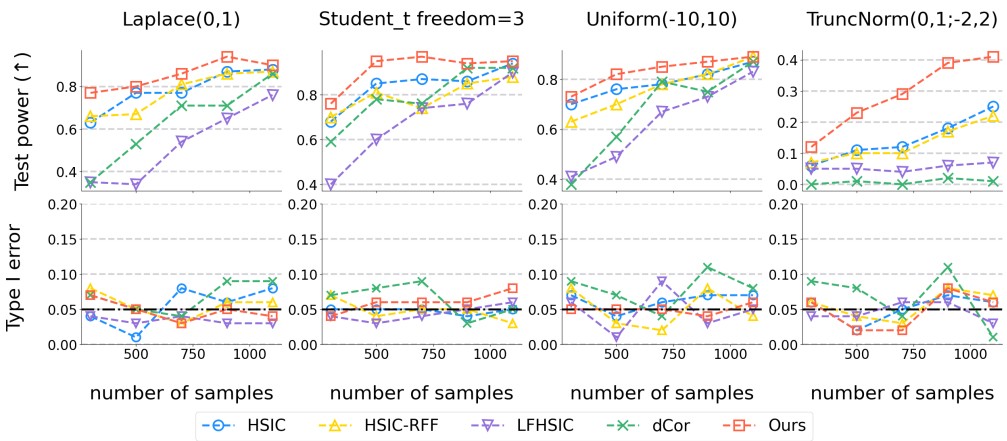

Figure 4: The experiment results when we change the sample sizes of the linear mixtures of 3 independent components from different distributions. The sample sizes $n \in \{300, 500, 700, 900, 1100\}$.

**Results.** In Figure 3, we demonstrate that our method consistently controls Type I errors and that its power outperforms other baselines in different distributions. More independent and identically distributed components make the standardized linear mixture behaves more like Gaussian[5], and the dependence relation between $X$ and $Y$ becomes more complex and hard to detect. Note that our method does not suffer from this complexity as much as other baselines do for Laplace$(0,1)$ and $t_{\nu=3}$. Figure 4 shows the performance when the number of samples varies. All methods benefit from a power gain when more data samples are available. Our method again consistently performs better in testing power, which confirms the need for a specific testing method for linear non-Gaussian data.

## 6.2 APPLICATION IN CAUSAL DISCOVERY METHODS

We then apply our method to classical causal discovery method with linear non-Gaussian assumption. Here we use the ground truth causal structure of the flow cytometry dataset, SACHS (Sachs et al., 2005), and test the algorithm on both original real data and synthetic data generated according to the structure. We report the Structural Hamming Distance (SHD) and F1 score. SHD measures how many edge insertions, deletions, or reversals are needed to transform the estimated graph into the ground-truth graph. F1 score is the harmonic mean of precision and recall, summarizing how accurately and completely the estimated edges recover the true causal edges. More details about the dataset SACHS and results of the real data see Appendix E.4 and E.5.

**Direct-LiNGAM.** This is a causal discovery algorithm for LiNGAM without latent confounders. It finds the causal order by repeatedly identifying exogenous variables via independence tests with regression residuals. We replace the original HSIC test with other baselines and our method, and run the algorithm on the synthetic data we generated according to the structure of SACHS. We fix the sample size as $n = 500$ and change the distribution of the exogenous variables. We can find the consistently better performance of our LiNGIC in all settings as shown in Table 1.

---

[5]With assumption that these components have finite variance.

Table 1: SHD and F1 Score of Direct-LiNGAM algorithm using different testing methods.

| Noise Type | SHD (↓) | | | | F1 Score (↑) | | | |
|---|---|---|---|---|---|---|---|---|
| | HSIC | HSIC-RFF | dCor | Ours | HSIC | HSIC-RFF | dCor | Ours |
| *Uniform* | 2.1 | 2.8 | 0.9 | **0.8** | 0.93 | 0.91 | 0.97 | **0.98** |
| *Laplace* | 1 | 2.7 | 4.7 | **0.2** | 0.96 | 0.91 | 0.84 | **0.99** |
| *Student t* | 2.6 | 2.6 | 2.5 | **2.3** | 0.91 | 0.90 | 0.91 | 0.91 |
| *TruncNorm* | 13.2 | 13.9 | 17.1 | **6.3** | 0.60 | 0.55 | 0.45 | **0.79** |

## 7 CONCLUSION AND DISCUSSION

We studied the problem of testing independence between linear mixtures of independent non-Gaussian components, a critical but underexplored task in machine learning and causal discovery. We established a new theoretical characterization showing that independence is fully determined by the constancy of the conditional mean and variance under this setting. Building on this insight, we proposed a kernel-based testing procedure with provable asymptotic guarantees. Experiments demonstrate clear power gains by leveraging the model constraints. Overall, our findings provide both theoretical and practical advances for causal discovery methods with linear non-Gaussian assumption and highlight the importance of exploiting structural assumptions in data.

## ACKNOWLEDGMENTS

The authors would like to thank the anonymous reviewers for their helpful comments. We would also like to acknowledge the support from NSF Award No. 2229881, AI Institute for Societal Decision Making (AI-SDM), the National Institutes of Health (NIH) under Contract R01HL159805, and grants from Quris AI, Florin Court Capital, MBZUAI-WIS Joint Program, and the Al Deira Causal Education project. XW also acknowledges the support from National Natural Science Foundation of China (Grants No. 12171076).

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

APPENDIX

**Organization of Appendices**

- Section A: Table of Symbols and Notations.

- Section B: Use of the LLM.

- Section C: The Proof of the Main Results.

- Section D: The Derivation of Statistics.

- Section E: Supplementary Experimental Details and Results.

- Section F: Discussions.

## A    NOTATIONS

Table 2: Notation Table

| Symbol | Description |
|---|---|
| $X, Y$ | Random variables (or sets of variables) |
| $\mathcal{X}, \mathcal{Y}$ | Domains for random variables |
| $\mathcal{F}_X, \mathcal{F}_Y$ | Reproducing kernel Hilbert spaces (RKHS) |
| $\mathbf{x}, \mathbf{y}$ | Sample vectors (or matrices) |
| $x_i, y_i, z_i$ | Specific values of sample vectors (or matrices) |
| $k_X(x, x'), k_Y(y, y')$ | Kernel functions on the input spaces $\mathcal{X}, \mathcal{Y}$ |
| $\psi(\cdot), \phi(\cdot)$ | Feature maps for $X, Y$ |
| $\boldsymbol{K}_X, \boldsymbol{K}_Y$ | Kernel matrice on samples $x, y$ |
| $\Sigma_{XY}$ | Cross-Covariance operator |
| $\| \cdot \|_{\mathcal{F}}$ | Norm in a RKHS |
| $\mathbb{E}[X]$ | Expectation of $X$ |
| $\mathbb{V}\mathrm{ar}[X]$ | Variance of $X$ |
| $\mathbb{C}\mathrm{ov}[X]$ | Covariance of $X$ and $Y$ |
| $\mathbb{R}^{\geq 0}$ | The set of positive real numbers (including 0) |
| $\mathcal{B}(\mathbb{R})$ | Borel $\sigma$-algebra on $\mathbb{R}$ |
| $\mathbb{P}_{XY}$ | Joint distribution of $X$ and $Y$ |
| $\mathbb{P}_{XY|Z}$ | Joint distribution of $X$ and $Y$ conditioned on $Z$ |
| $\mathrm{Tr}[\cdot]$ or $\mathrm{trace}(\cdot)$ | The trace of a matrix |
| $\otimes$ | Tensor product |
| $\mathcal{O}$ | Big O notion |
| $n$ | Number of samples |
| $X \perp\!\!\!\perp Y$ | $X$ is independent of $Y$ |
| $(i)_r^n$ | The set of all $r$-tuples drawn without replacement |
| $(n)_k$ | Number of permutations |
| $\mathcal{N}(0, 1)$ | Normal distribution with zero mean and standard deviation 1 |
| $\mathcal{U}(0, 1)$ | Uniform distribution in $(0, 1)$ |

## B    THE USE OF LARGE LANGUAGE MODELS

During the preparation of this manuscript, we used ChatGPT to assist with writing by providing the prompt: "Please check whether this part is suitable for a paper for submission to an international conference." We applied its suggestions paragraph by paragraph, and all outputs were edited by us to ensure correctness.

## C PROOF OF THE MAIN RESULTS

### C.1 PRELIMINARIES AND USEFUL LEMMAS

**Theorem C.1** (Cram´er Decomposition Theorem (Cramér, 1970))**.** *Let a random variable $\varepsilon$ be normally distributed and admit a decomposition as a sum $\varepsilon = \varepsilon_1 + \varepsilon_2$ of two independent random variables. Then the summands $\varepsilon_1$ and $\varepsilon_2$ are normally distributed as well.*

**Theorem C.2** (Darmois-Skitovich Theorem (DST))**.** *We consider independent scalar random variables $X_1, \ldots, X_n$ (not necessarily identically distributed) and two linear statistics*

$$L_1 = \sum \alpha_i X_i, \quad L_2 = \sum \beta_i X_i,$$

*where the $\alpha_i, \beta_i$ are constant coefficients. Let $L_1$ and $L_2$ be independent. Then the random variables $X_j$ for which $\alpha_j \beta_j \neq 0$ are all normal.*

As mentioned in (Shimizu et al., 2011) In other words, this theorem means that if there exists a non-Gaussian $X_j$ for which $\alpha_j \beta_j \neq 0$, $L_1$ and $L_2$ are dependent.

The following lemmas and corresponding proofs are attributed to (Kagan et al., 1973).

**Lemma C.3.** *Let $X$ and $Y$ be random variables and $EY$ exist. $Y$ has constant regression on $X$ if and only if the relation*

$$E\left(Y e^{itX}\right) = EY \cdot E e^{itX},$$

*holds for all real $t$.*

**Lemma C.4.** *Let $(X, Y)$ be a two-dimensional random vector with $EX = EY = 0$. A necessary and sufficient condition for the linearity of the regression of $Y$ on $X$ is the existence of a constant $\beta$ such that, for all real $t_1$,*

$$\left. \frac{\partial \phi(t_1, t_2)}{\partial t_2} \right|_{t_2=0} = \beta \frac{d\phi(t_1, 0)}{dt_1},$$

*where $\phi(t_1, t_2)$ is the c.f. of $(X, Y)$.*

*Proof.* In view of Lemma 1.1.1, a necessary and sufficient condition for $E(Y - \beta X \mid X) = 0$ for some constant $\beta$ is that

$$E(Y - \beta X)e^{itX} = 0 \quad \text{for all real } t,$$

which is easily seen to reduce to the results here. □

**Lemma C.5.** *In order for the two-dimensional random vector $(X, Y)$ to satisfy the conditions*

$$E(Y \mid X) = \alpha + \beta X,$$
$$\mathrm{Var}(Y \mid X) = \sigma^2 = \text{constant},$$

*it is necessary and sufficient that*

$$\left. \begin{array}{l} \left. \frac{\partial \phi(t_1, t_2)}{\partial t_2} \right|_{t_2=0} = i\alpha \phi(t_1, 0) + \beta \frac{d\phi(t_1, 0)}{dt_1} \\[2mm] \left. \frac{\partial^2 \phi(t_1, t_2)}{\partial t_2^2} \right|_{t_2=0} = -\left(\sigma^2 + \alpha^2\right) \phi(t_1, 0) + 2i\alpha\beta \frac{d\phi(t_1, 0)}{dt_1} + \beta^2 \frac{d^2\phi(t_1, 0)}{dt_1^2} \end{array} \right\}. \tag{5}$$

*Proof.* The conditions here are equivalent to

$$E(Y - \beta X \mid X) = \alpha,$$
$$E\left[Y^2 - (\alpha + \beta X)^2 \mid X\right] = \sigma^2.$$

which, in view of Lemma C.3, are easily seen to be equivalent to the results in the Lemma. □

**Lemma C.6** ($\alpha$-decomposition theorem)**.** *Let the function $\phi(z)$ of the complex variable $z$ be regular and nonvanishing on the disc $|z| < R$ and possess the Hermitian property: $\phi(-z) = \overline{\phi(\bar{z})}$. If $\phi_1, \ldots, \phi_s$ be c.f.s, and $\alpha_1, \ldots, \alpha_s$ be positive numbers such that for some sequence $\{t_n\}$ of real numbers tending to zero the relation*

$$[\phi_1(t)]^{\alpha_1} \cdots [\phi_s(t)]^{\alpha_s} = \phi(t) \tag{6}$$

*is satisfied, then the functions $\phi_j$ are regular and nonvanishing in $|z| < R$, and relation (6) is valid throughout the disc. If in (6), $\phi$ is a function of the form $\exp Q(t)$, where $Q(t)$ is a polynomial with the Hermitian property, then every $\phi_j$ is a normal c.f.*

## C.2 A PRELIMINARY RESULTS AND THE CORRECTION OF ITS PROOF

**Theorem C.7** (THEOREM 5.7.1. in (Kagan et al., 1973)). *Let $X_1, \ldots, X_n$ be independent r.v.'s with finite variance. The linear functions $L = \sum a_j X_j$ and $M = \sum b_j X_j$ with $a_j b_j \neq 0$ for $j = 1, \ldots, n$ satisfy the relations*

*(i) $E(L \mid M) = \alpha + \beta M$, and*

*(ii) $\mathrm{Var}(L \mid M) = \sigma_0^2 = constant$*

*if and only if the following conditions are satisfied:*

*(a) the $X_j$ for which $a_j \neq \beta b_j$ are normal, and*

*(b) $\beta = \left( \sum{}^* a_j b_j \sigma_j^2 \right) / \left( \sum{}^* b_j^2 \sigma_j^2 \right), \quad \sigma_0^2 = \sum{}^* (a_j - \beta b_j)^2 \sigma_j^2,$*

*where $\sigma_j^2 = \mathrm{Var}\, X_j$, and $\sum{}^*$ denotes that the summation is taken over all $j$ for which $a_j \neq \beta b_j$.*

We claim that the original proof has minor problem with an exchanged $a$ and $b$. Here we give another proof that corrects this problem.

*Proof.* The sufficiency of the conditions is easily established.

$$\log \phi (t_1, t_2) = \log E \left[ e^{i(t_1 M + t_2 L)} \right] = \log \left[ e^{i \sum_{j=1}^n (b_j t_1 + a_j t_2) \varepsilon_j} \right]$$

$$= \log \left[ \prod_{j=1}^n f_j (b_j t_1 + a_j t_2) \right] = \sum_{j=1}^n \theta_j (t_1, t_2).$$

$$\phi (t_1, t_2) = e^\theta \Rightarrow \begin{cases} \phi' = \theta' \phi \\ \phi'' = \theta'' \phi + \phi' \theta' = \phi \left( \theta'' + \theta^2 \right) \end{cases} .$$

We first prove the necessity. Let $f_j$ be the c.f. of $X_j$, and $\theta_j = \log f_j$ (in a neighborhood of the origin where none of the $f_j$ vanishes).

$$\sum a_j \theta_j' (b_j t) = i\alpha + \beta \sum b_j \theta_j' (b_j t), \tag{7}$$

$$\sum a_j^2 \theta_j'' (b_j t) = -\sigma_0^2 + \beta^2 \sum b_j^2 \theta_j'' (b_j t). \tag{8}$$

Differentiating (5.7.1) with respect to $t$, we obtain

$$\sum a_j b_j \theta_j'' (b_j t) = \beta \sum b_j^2 \theta_j'' (b_j t).$$

From (5.7.2) and (5.7.3), we derive

$$\sum (a_j - \beta b_j)^2 \theta_j'' (b_j t) = \sum{}^* (a_j - \beta b_j)^2 \theta_j'' (b_j t) = \sigma_0^2.$$

Integrating (5.7.4), we obtain

$$\prod{}^* [f_j (b_j t)]^{\gamma_j} = \exp \left[ i\mu t - (1/2)\sigma_0^2 t^2 \right],$$

where $\gamma_j = (a_j - \beta b_j)^2$. Assertion (a) now follows on noting that in Lemma C.6, if the above holds with $\phi(t) = \exp \left[ i\mu t - (1/2)\sigma^2 t^2 \right]$, then every $\phi_i$ is a normal c.f. The other results are obtained by setting $t = 0$ in (7) and (7).

The sufficiency of the conditions is easily established. $\qquad \square$

## C.3 PROOF OF THEOREM 4.2

**Theorem 4.2** *Let $\varepsilon_1, \ldots, \varepsilon_n$ be independent non-Gaussian r.v.'s with finite variance. Suppose that $Y = \sum a_j \varepsilon_j$ and $X = \sum b_j \varepsilon_j$ be two linear mixtures of $\varepsilon_i$. Then $Y \perp\!\!\!\perp X$ if and only if:*

*(i) $E(Y \mid X) = c =$constant a.s., and*

*(ii) $\mathrm{Var}(Y \mid X) = \sigma_0^2 =$ constant a.s.*

*Proof.* $\Rightarrow$: Suppose that we know $X \perp\!\!\!\perp Y$, then we directly have (i) and (ii).

$\Leftarrow$: Proof by contradiction. Suppose that we have conditions (i) and (ii). We can rewrite $X$ and $Y$ in

$$Y = L + A, \ L = \sum_{i \in \mathcal{S}} a_i \varepsilon_i, A = \sum_{i \in \mathcal{S}^c} a_i \varepsilon_i, \quad X = M + B, \ M = \sum_{i \in \mathcal{S}} b_i \varepsilon_i, B = \sum_{i \in \mathcal{S}^c} b_i \varepsilon_i,$$

where $\mathcal{S} = \{i \mid a_i b_i \neq 0\}$ and $\mathcal{S}^c \cup \mathcal{S} = \{1, ..., n\}$. Since $\varepsilon_i, \forall i$ are mutually independent, the linear mixtures of disjoint sets of $\varepsilon_i$s are independent, i.e., $A \perp\!\!\!\perp B, L \perp\!\!\!\perp A, B$, and $M \perp\!\!\!\perp A, B$. Then we have

$$\mathbb{E}[Y \mid X] = \mathbb{E}[L + A \mid X] = \mathbb{E}[L \mid X] + \mathbb{E}[A \mid X] = c \Rightarrow \mathbb{E}[L \mid X] = \alpha \triangleq c - \mathbb{E}[A], \quad (9)$$

$$\mathbb{V}ar(Y \mid X) = \mathbb{V}ar(L + A \mid X) = \mathbb{V}ar(L \mid X) + \mathbb{V}ar(A), \quad (10)$$

$$\Rightarrow \mathbb{V}ar(L \mid X) = \sigma^2 \triangleq \sigma_0^2 - \mathbb{V}ar(A). \quad (11)$$

We assign $L = \sum_{j=1}^n a_j^{(l)} \varepsilon_j = \sum_{j \in \mathcal{S}} a_j \varepsilon_j$, where $a_j^{(l)} = 0, \forall j \in \mathcal{S}^c$. Let $f_j$ be the characteristic function (c.f.) of $\varepsilon_j$, and $\theta_j = \log f_j$ (in a neighborhood of the origin where none of the $f_j$ vanishes). The c.f. of $(X, L)$ is $\phi(t_1, t_2) = \mathbb{E}[e^{i(t_1 M + t_2 L)}]$. Then through Lemma 4.1 we know $\beta = 0$,

$$\sum_{j=1}^n a_j^{(l)} \theta_j'(b_j t) = \sum_{j \in \mathcal{S}} a_j \theta_j'(b_j t) = i\alpha, \quad (12)$$

$$\sum_{j=1}^n a_j^{(l) \, 2} \theta_j''(b_j t) = \sum_{j \in \mathcal{S}} a_j^2 \theta_j''(b_j t) = -\sigma^2. \quad (13)$$

So we only need to consider $j \in \mathcal{S} = \{i \mid a_i b_i \neq 0\}$. Also we know that for $j \in \mathcal{S}$, we have $b_j \neq 0$. We now want to integrate (13). Suppose $F(t) = \sum_{j \in \mathcal{S}} a_j^2 \frac{1}{b_j} \theta_j'(b_j t)$ since $b_j \neq 0$ for $j \in \mathcal{S}$. Then $F'(t) = \sum_{j \in \mathcal{S}} a_j^2 \theta_j''(b_j t) = -\sigma^2$. Therefore $F(t) = -\sigma^2 t + C_1$. Suppose again $G(t) = \sum_{j \in \mathcal{S}} a_j^2 \frac{1}{b_j^2} \theta_j(b_j t)$. Then $G'(t) = \sum_{j \in \mathcal{S}} a_j^2 \frac{1}{b_j} \theta_j'(b_j t) = F(t) = -\sigma^2 t + C_1$, which gives us $G(t) = -\frac{1}{2}\sigma^2 t^2 + C_1 t + C_0$. Then determine the constants using the initial values.

$$\theta_j(b_j t)\big|_{t=0} = \log f_j(0) = \log \mathbb{E}[e^{i\varepsilon_j \cdot 0}] = 0, \quad (14)$$

$$\theta_j'(b_j t)\big|_{t=0} = \frac{f_j'(b_j t)\big|_{t=0}}{f_j(0)} = \frac{\mathbb{E}[i\varepsilon_j \cdot e^{i\varepsilon_j \cdot 0}]}{\mathbb{E}[e^{i\varepsilon_j \cdot 0}]} = \mathbb{E}[i\varepsilon_j]. \quad (15)$$

These give us the following points

$$G(0) = \sum_{j \in \mathcal{S}} a_j^2 \frac{1}{b_j^2} \theta_j(b_j \cdot 0) = 0, \quad (16)$$

$$G'(0) = F(0) = \sum_{j \in \mathcal{S}} a_j^2 \frac{1}{b_j} \theta_j'(b_j \cdot 0) = \sum_{j \in \mathcal{S}} a_j^2 \frac{1}{b_j} \mathbb{E}[i\varepsilon_j] \triangleq i\mu. \quad (17)$$

Therefore, the integrated function is $G(t) = \sum_{j \in \mathcal{S}} a_j^2 \frac{1}{b_j^2} \theta_j(b_j t) = -\frac{1}{2}\sigma^2 t^2 + i\mu t$. We take exponential on both sides,

$$\exp(i\mu t - \frac{1}{2}\sigma^2 t^2) = \Pi_{j \in \mathcal{S}} \left[ f_j(b_j t) \right]^{a_j^2 / b_j^2}.$$

Since $i\mu t - \frac{1}{2}\sigma^2 t^2$ has the Hermitian property, through Lemma C.6, all $f_j$s are normal c.f. and $\varepsilon_j$s are normal variables, which contradicts to the linear non-Gaussian model. So $\mathcal{S} = \emptyset$.

So now we have $Y = A = \sum_{i \in \mathcal{A}} a_i \varepsilon_i$ and $X = B = \sum_{i \in \mathcal{B}} b_i \varepsilon_i$. Since $\mathcal{A} \cap \mathcal{B} = \emptyset$ and $\varepsilon_i$s are mutually independent, we have $X \perp\!\!\!\perp Y$. $\qquad\square$

## C.4 PROOF OF THEOREM 4.3

**Lemma C.8.** *Let $(\Omega, \mathcal{F}, \mathbb{P})$ be a probability space, $X : \Omega \to \mathcal{X}$ a random element taking values in a metric space $\mathcal{X}$, and let $h : \mathcal{X} \to \mathbb{R}$ be Borel measurable with $\mathbb{E}|h(X)| < \infty$. Assume that*

$$\mathbb{E}\big[f(X)\,h(X)\big] = 0 \qquad \text{for all } f \in C_b(\mathcal{X}).$$

*Then $h(X) = 0$ almost surely. Moreover, if $h$ is continuous and the law of $X$ has full support on $\mathcal{X}$ (i.e., $\mathbb{P}(X \in U) > 0$ for every nonempty open set $U$), then $h(x) \equiv 0$ for all $x \in \mathcal{X}$.*

*Proof. Remark:* This result can be viewed as a measure-theoretic analogue of the fundamental lemma of calculus of variations (Gelfand et al., 2000). We provide a self-contained proof below for completeness through contradiction. Let $\mu = \mathbb{P} \circ X^{-1}$ be the law of $X$.

*1) Indicator approximation.* By regularity of Borel probability measures on metric spaces and Urysohn's Lemma, for every Borel set $A \subset \mathcal{X}$ there exists $(f_n) \subset C_b(\mathcal{X})$ with $0 \le f_n \le 1$ and $f_n \to \mathbf{1}_A$ $\mu$-a.e., where $\mathbf{1}_A$ represents the indicator function of the set $A$.

*2) Extension to bounded measurable functions.* For a simple function $g = \sum_{i=1}^m c_i \mathbf{1}_{A_i}$, define $g_n = \sum_{i=1}^m c_i f_n^{(i)} \in C_b(\mathcal{X})$ using the above approximations. Then $g_n(X) \to g(X)$ a.s. and $|g_n(X)h(X)| \le \left(\sum_i |c_i|\right)|h(X)|$ with $\mathbb{E}|h(X)| < \infty$. By the Dominated Convergence Theorem,

$$\mathbb{E}[g(X)h(X)] = \lim_{n \to \infty} \mathbb{E}[g_n(X)h(X)] = 0.$$

By approximation of bounded measurable functions by simple functions, the identity $\mathbb{E}[g(X)h(X)] = 0$ holds for every bounded Borel measurable $g$. Taking $g = \mathbf{1}_A$ yields $\int_A h\,d\mu = 0$ for every Borel $A$.

*3) Conclusion.* If $\mathbb{P}(h(X) \ge \varepsilon) > 0$ for some $\varepsilon > 0$, then $0 = \int_{\{h \ge \varepsilon\}} h\,d\mu \ge \varepsilon\,\mu(\{h \ge \varepsilon\}) > 0$, a contradiction. Similarly for $\{h \le -\varepsilon\}$. Hence $h(X) = 0$ a.s.

If, in addition, $h$ is continuous and the law of $X$ has full support, then $h(x_0) \ne 0$ would imply $|h| \ge c > 0$ on an open ball around $x_0$, which has positive probability. This contradicts $h(X) = 0$ a.s., and therefore $h \equiv 0$ everywhere. $\qquad\square$

**Theorem 4.3** (Independence for Linear Non-Gaussian Data). *The linear mixtures of the independent non-Gaussian variables $\varepsilon_1, \dots, \varepsilon_m$, $Y = \sum_{i=1}^n a_i \varepsilon_i$ and $X = \sum_{i=1}^n b_i \varepsilon_i$, are independent if and only if $\mathrm{Cov}(f(X), Y) = 0$ and $\mathrm{Cov}(f(X), Y^2) = 0$, where $f$ can be any bounded, continuous function.*

*Proof.* $\Rightarrow$ If $X \perp\!\!\!\perp Y$, then from Lemma 2.1, we know $\mathrm{Cov}(f(X), g(Y)) = 0$ for each pair $(f, g)$ of bounded, continuous functions. Clearly the condition is satisfied.

$\Leftarrow$ We first consider $\mathrm{Cov}(f(X), Y) = 0$. For any bounded and continuous function $f$,

$$\mathrm{Cov}(f(X), Y) = \mathbb{E}[f(X)Y] - \mathbb{E}[f(X)]\mathbb{E}[Y] = \mathbb{E}[f(X)(Y - \mathbb{E}[Y])]$$
$$= \mathbb{E}[f(X)\mathbb{E}[(Y - \mathbb{E}[Y])|X]] = 0.$$

If we define $h(X) \triangleq \mathbb{E}[(Y - \mathbb{E}Y)|X] = \mathbb{E}(Y \mid X) - \mathbb{E}[Y]$, then

$$\mathbb{E}[f(X)h(X)] = 0, \quad \forall f \in \mathcal{C}_b(X).$$

Use the results in Lemma C.8, we get that $h(X) = 0$ a.s. Therefore we have the condition $\mathbb{E}(Y \mid X) = \mathbb{E}[Y]$ a.s., which is a constant.

With a similar discussion, since $\mathrm{Cov}(f(X), Y^2) = 0$ gives us $\mathbb{E}[Y^2|X] = \mathbb{E}[Y^2]$ a.s. and it is a constant, we can derive that

$$\mathbb{V}ar(Y \mid X) = \mathbb{E}[Y^2 - \mathbb{E}[Y]^2 \mid X] = \mathbb{E}[Y^2|X] - \mathbb{E}[Y]^2 \triangleq c_2 = \text{constant}.$$

Combine the results and use Theorem 4.2, we know that $X \perp\!\!\!\perp Y$. $\qquad\square$

# D DETAILS ABOUT THE UNCONDITIONAL INDEPENDENCE STATISTIC

We first give some preliminaries for later proof and derivation.

**Definition D.1** ($U$-statistics). The statistic $U_n$ defined as follows is called a $U$-statistic with symmetric function $h$ of order $m$:

$$U_n = \binom{n}{m}^{-1} \sum_c h\left(X_{i_1}, \ldots, X_{i_m}\right), \tag{18}$$

where $\sum_c$ denotes the summation over the $\binom{n}{m}$ combinations of $m$ distinct elements $\{i_1, \ldots, i_m\}$ from $\{1, \ldots, n\}$.

For every $U$-statistic $U_n$ as an estimator of $\vartheta = E\left[h\left(X_1, \ldots, X_m\right)\right]$, there is a closely related $V$-statistic defined by

$$V_n = \frac{1}{n^m} \sum_{i_1=1}^n \cdots \sum_{i_m=1}^n h\left(X_{i_1}, \ldots, X_{i_m}\right).$$

**Proposition D.2.** *Let $V_n$ be defined by the above function and we have $n$ i.i.d. samples $\{x_i\}_{i=1}^n$ drawn from $\mathbb{P}_X$.*

*(i) Assume that $\mathbb{E}[|h\left(X_{i_1}, \ldots, X_{i_m}\right)|] < \infty$ for all $1 \le i_1 \le \cdots \le i_m \le m$. Then the bias of $V_n$ satisfies*

$$b_{V_n}(\mathbb{P}_X) = O\left(n^{-1}\right).$$

*(ii) Assume that $\mathbb{E}\left[h\left(X_{i_1}, \ldots, X_{i_m}\right)^2\right] < \infty$ for all $1 \le i_1 \le \cdots \le i_m \le m$. Then the variance of $V_n$ satisfies*

$$\mathbb{V}\mathrm{ar}(V_n) = \mathbb{V}\mathrm{ar}(U_n) + O(n^{-2}).$$

We also define some more statistics for later derivation. For $k = 1, \ldots, m$, let

$$\begin{aligned} h_k(x_1, \ldots, x_k) &= \mathbb{E}\left[h(X_1, \ldots, X_m) \mid X_1 = x_1, \ldots, X_k = x_k\right] \\ &= \mathbb{E}\left[h(x_1, \ldots, x_k, X_{k+1}, \ldots, X_m)\right]. \end{aligned}$$

Note that $h_m = h$. Further define $\zeta_k \triangleq \mathbb{V}\mathrm{ar}\left(h_k(X_1, \ldots, X_k)\right)$.

**Theorem D.3** ((Shao, 2008), Theorem 3.16). *Let $V_n$ be a V-statistics with $\mathbb{E}\left[h(X_{i_1}, \ldots, X_{i_m})^2\right] < \infty$ for all $1 \le i_1 \le \cdots \le i_m \le m$.*

*(i) If $\zeta_1 \triangleq \mathbb{V}\mathrm{ar}(h_1(X_1)) > 0$, then*

$$\sqrt{n}\left(V_n - \vartheta\right) \xrightarrow{d} N(0, m^2 \zeta_1).$$

*(ii) If $\zeta_1 = 0$ but $\zeta_2 \triangleq \mathbb{V}\mathrm{ar}(h_2(X_1, X_2)) > 0$, then*

$$n\left(V_n - \vartheta\right) \xrightarrow{d} \frac{m(m-1)}{2} \sum_{j=1}^\infty \lambda_j \chi_{1j}^2,$$

*where $\chi_{1j}^2$ 's are i.i.d. random variables having the chi-square distribution $\chi_1^2$ and $\lambda_j$ 's are some constants (which may depend on $\mathbb{P}_X$ ) satisfying $\sum_{j=1}^\infty \lambda_j^2 = \zeta_2$.*

## D.1 CHARACTERIZATION OF UNCONDITIONAL INDEPENDENCE

For $(X, Y) \in \mathcal{X} \times \mathcal{Y}$, the cross-covariance operator $\Sigma_{XY} : \mathcal{F}_Y \to \mathcal{F}_X$ is defined by (Fukumizu et al., 2004):

$$\forall f \in \mathcal{F}_X, g \in \mathcal{F}_Y, \langle f, \Sigma_{XY} g \rangle_{\mathcal{F}_X} = \mathbb{E}_{XY}[f(X)g(Y)] - \mathbb{E}_X[f(X)]\mathbb{E}_Y[g(Y)]. \tag{19}$$

and the covariance operator itself can be written as

$$\Sigma_{XY} := \mathbb{E}_{XY}\left[(\psi(X) - \mu_X) \otimes (\phi(Y) - \mu_Y)\right], \ \mu_X \triangleq \mathbb{E}_X\psi(X), \mu_Y \triangleq \mathbb{E}_Y\phi(Y), \qquad (20)$$

where $\otimes$ is the tensor product. This operator is a generalization of the cross-covariance matrix between random vectors. HSIC is the squared Hilbert-Schmidt norm (the sum of the squared singular values) of this operator, as mentioned in Def. 2.3

$$\text{HSIC}(X, Y) = \mathbb{E}_{XX'YY'}[k_X(X, X')k_Y(Y, Y')] + \mathbb{E}_{XX'}[k_X(X, X')]\mathbb{E}_{YY'}[k_Y(Y, Y')] \quad (21)$$
$$- 2\mathbb{E}_{XY}\left[\mathbb{E}_{X'}[k_X(X, X')]\mathbb{E}_{Y'}[k_Y(Y, Y')]\right].$$

Assuming the expectations exist, where $X'$ denotes an independent copy of $X$. An unbiased estimator of HSIC in sample $\mathcal{D} = \{(x_i, y_i)\}_{i=1}^n$ drawn from distribution $\mathbb{P}_{XY}$ is the sum of three $U$-statistics: (Gretton et al., 2007)

$$\text{HSIC}_u(\mathcal{D}) = \frac{1}{(n)_2} \sum_{(i,j)\in\mathbf{i}_2^n} k_X^{ij}k_Y^{ij} + \frac{1}{(n)_4} \sum_{(i,j,q,r)\in\mathbf{i}_4^n} k_X^{ij}k_Y^{qr} - 2\frac{1}{(n)_3} \sum_{(i,j,q)\in\mathbf{i}_3^n} k_X^{ij}k_Y^{iq}, \qquad (22)$$

where $k_X^{ij} := k_X(x_i, x_j)$, $k_Y^{ij} := k_Y(y_i, y_j)$, $(n)_m := \frac{n!}{(n-m)!}$, and the index set $\mathbf{i}_r^n$ denotes the set all $r$-tuples drawn without replacement from the set $\{1, \ldots, n\}$. A biased estimator is the one replacing $U$-statistics with $V$-statistics, as in

$$\text{HSIC}_b(\mathcal{D}) = \frac{1}{n^2} \sum_{i,j}^n k_X^{ij}k_Y^{ij} + \frac{1}{n^4} \sum_{i,j,q,r}^n k_X^{ij}k_Y^{qr} - 2\frac{1}{n^3} \sum_{i,j,q}^n k_X^{ij}k_Y^{iq} = \frac{1}{n^2} \text{Tr}(\boldsymbol{K}_X\boldsymbol{H}\boldsymbol{K}_Y\boldsymbol{H}), \quad (23)$$

where the summation indices now denote all $r$-tuples drawn with replacement from $\{1, \ldots, n\}$ and $\boldsymbol{H} = \boldsymbol{I} - \frac{1}{n}\mathbf{1}\mathbf{1}^\top$.

## D.2 Approximate the asymptotic null distribution

### D.2.1 Mean of $\text{LiNGIC}_b(\mathcal{D})$ under $\mathcal{H}_0$

An unbiased estimate of $\text{LiNGIC}(X, Y)$, denoted by $\text{LiNGIC}_u(\mathcal{D})$, is a sum of three U-statistics

$$\text{LiNGIC}_{1u}(\mathcal{D}) := \frac{1}{(n)_2} \sum_{(i,j)\in\mathbf{i}_2^n} k_{ij}l_{ij} + \frac{1}{(n)_4} \sum_{(i,j,q,r)\in\mathbf{i}_4^n} k_{ij}l_{qr} - 2\frac{1}{(n)_3} \sum_{(i,j,q)\in\mathbf{i}_3^n} k_{ij}l_{iq},$$

which has $\mathbb{E}\left[\text{LiNGIC}_{1u}(\mathcal{D})\right] = \mathbb{E}[\text{LiNGIC}(X, Y)] = 0$ under $\mathcal{H}_0$.

The complete proof is given in (Gretton et al., 2007). We show only some of the key steps here. The biased estimate of $\text{LiNGIC}(X, Y)$, denote as $\text{LiNGIC}_b(\mathcal{D})$, is a sum of three V-statistics

$$\text{LiNGIC}_{1b}(Z) := \frac{1}{n^2} \sum_{i,j}^n k_{ij}l_{ij} + \frac{1}{n^4} \sum_{i,j,q,r}^n k_{ij}l_{qr} - 2\frac{1}{n^3} \sum_{i,j,q}^n k_{ij}l_{iq}$$

We can show the difference $n\left(\text{LiNGIC}_b(Z) - \text{LiNGIC}_u(Z)\right)$ similar to (Ren et al., 2024):

$$
\begin{aligned}
=& \frac{1}{n}\sum_i k_{ii}^X l_{ii}^Y - \frac{2}{n^2} \sum_{(i,j)\in\mathbf{i}_2^n} \left(k_{ii}^X l_{ij}^Y + k_{ij}^X l_{ii}^Y\right) + \frac{1}{n^3} \sum_{(i,j,q)\in\mathbf{i}_3^n} \left(k_{ii}^X l_{jq}^Y + k_{ij}^X l_{qq}^Y\right) \\
& - \frac{3}{(n)_2} \sum_{(i,j)\in\mathbf{i}_2^n} k_{ij}^X l_{ij}^Y + \frac{10}{(n)_3} \sum_{(i,j,q)\in\mathbf{i}_3^n} k_{ij}^X l_{iq}^Y - \frac{6}{(n)_4} \sum_{(i,j,q,r)\in\mathbf{i}_4^n} k_{ij}^X l_{qr}^Y \\
& + \frac{1}{n}\sum_i k_{ii}^Y l_{ii}^X - \frac{2}{n^2} \sum_{(i,j)\in\mathbf{i}_2^n} \left(k_{ii}^Y l_{ij}^X + k_{ij}^Y l_{ii}^X\right) + \frac{1}{n^3} \sum_{(i,j,q)\in\mathbf{i}_3^n} \left(k_{ii}^Y l_{jq}^X + k_{ij}^Y l_{qq}^X\right) \\
& - \frac{3}{(n)_2} \sum_{(i,j)\in\mathbf{i}_2^n} k_{ij}^Y l_{ij}^X + \frac{10}{(n)_3} \sum_{(i,j,q)\in\mathbf{i}_3^n} k_{ij}^Y l_{iq}^X - \frac{6}{(n)_4} \sum_{(i,j,q,r)\in\mathbf{i}_4^n} k_{ij}^Y l_{qr}^X + \mathcal{O}\left(n^{-1}\right),
\end{aligned}
$$

when we assume the kernel is bounded with compact $\mathcal{X}$ and $\mathcal{Y}$. Secondly, we take the expectation of the last equation. To simplify, we use the notation $\mathbb{E}_{xyy'}kl = \mathbb{E}_{xyy'}k(x,x)l(y,y')$ (and so on for the rest), then $n\left(\mathbb{E}\left[\mathrm{LiNGIC}_b(Z)\right] - \mathbb{E}\left[\mathrm{LiNGIC}_u(Z)\right]\right) =$

$$
\begin{aligned}
=&\mathbb{E}_{yx}kl - 2\left(\mathbb{E}_{yxx'}kl + \mathbb{E}_{yy'x}kl\right) + \mathbb{E}_{yx'x''}kl + \mathbb{E}_{yy'x''}kl \\
&- 3\mathbb{E}_{yy'xx'}kl + 10\mathbb{E}_{yy''xx'}kl - 6\mathbb{E}_{yy'}k\mathbb{E}_{xx'}l \\
.&+ \mathbb{E}_{xy}kl - 2\left(\mathbb{E}_{xyy'}kl + \mathbb{E}_{xx'y}kl\right) + \mathbb{E}_{xy'y''}kl + \mathbb{E}_{xx'y''}kl \\
&- 3\mathbb{E}_{xx'yy'}kl + 10\mathbb{E}_{xx'yy''}kl - 6\mathbb{E}_{xx'}k\mathbb{E}_{yy'}l + \mathcal{O}\left(n^{-1}\right)
\end{aligned}
$$

Under $\mathcal{H}_0, x$ is independent with $y$, thus we can draw the conclusions that $\mathbb{E}_{xyy'}kl = \mathbb{E}_{xy'y''}kl, \mathbb{E}_{xx'y}kl = \mathbb{E}_{xx'y''}kl$ and $\mathbb{E}_{xx'yy'}kl = \mathbb{E}_{xx'yy''}kl = \mathbb{E}_{xx'}k\mathbb{E}_{yy'}l$. Similarly, $\mathbb{E}_{yxx'}kl = \mathbb{E}_{yx'x''}kl, \mathbb{E}_{yy'x}kl = \mathbb{E}_{yy'x''}kl$ and $\mathbb{E}_{yy'xx'}kl = \mathbb{E}_{yy'xx''}kl = \mathbb{E}_{yy'}k\mathbb{E}_{xx'}l$. Combining with $\mathbb{E}\left[\mathrm{LiNGIC}_u(Z)\right] = 0$, we obtain that

$$
\begin{aligned}
\mathbb{E}\left[\mathrm{LiNGIC}_b(Z)\right] = &\frac{1}{n}\left(\mathbb{E}_{xy}kl + \left\|\mu_x^k\right\|^2 \left\|\mu_y^l\right\|^2 - \mathbb{E}_x k \left\|\mu_y^l\right\|^2 - \mathbb{E}_y l \left\|\mu_x^k\right\|^2\right) \\
&+ \frac{1}{n}\left(\mathbb{E}_{yx}kl + \left\|\mu_y^k\right\|^2 \left\|\mu_x^l\right\|^2 - \mathbb{E}_y k \left\|\mu_x^l\right\|^2 - \mathbb{E}_x l \left\|\mu_y^k\right\|^2\right) + \mathcal{O}\left(n^{-2}\right),
\end{aligned}
$$

where $\mu_x^k := \mathbb{E}_x\phi_g(x), \mu_x^l := \mathbb{E}_x\phi_p(x)$, and for $\mu_y^k, \mu_y^l$ are similar. Also note that the estimators of $\mathbb{E}_x k$ can be written as $\widehat{\mathbb{E}_x k} = \widehat{\mathbb{E}_x k(x,x)} = \widehat{\mu_{xx}^k} = \frac{1}{n}\sum_i k_{ii}^X$, which is the same for $\widehat{\mathbb{E}_y k} = \widehat{\mu_{yy}^k} = \frac{1}{n}\sum_i k_{ii}^Y$, $\widehat{\mathbb{E}_x l} = \widehat{\mu_{xx}^l} = \frac{1}{n}\sum_i l_{ii}^X$, $\widehat{\mathbb{E}_y l} = \widehat{\mu_{yy}^l} = \frac{1}{n}\sum_i l_{ii}^Y$. An empirical estimate can be obtained by replacing the term above with

$$
\widehat{\left\|\mu_x^k\right\|^2} = \frac{1}{(n)_2}\sum_{(i,j)\in i_2^n} k_{ij}^X, \quad \widehat{\left\|\mu_y^l\right\|^2} = \frac{1}{(n)_2}\sum_{(i,j)\in \mathbf{i}_2^n} l_{ij}^Y, \quad \widehat{\left\|\mu_y^k\right\|^2} = \frac{1}{(n)_2}\sum_{(i,j)\in i_2^n} k_{ij}^Y,
$$

$$
\widehat{\left\|\mu_x^l\right\|^2} = \frac{1}{(n)_2}\sum_{(i,j)\in \mathbf{i}_2^n} l_{ij}^X.
$$

The obtained estimate

$$
\begin{aligned}
\mathbb{E}\left[n\,\mathrm{LiNGIC}_b(Z)\right] = &\widehat{\mu_{xx}^k\mu_{yy}^l} + \widehat{\left\|\mu_x^k\right\|^2}\widehat{\left\|\mu_y^l\right\|^2} - \widehat{\mu_{xx}^k}\widehat{\left\|\mu_y^l\right\|^2} - \widehat{\mu_{yy}^l}\widehat{\left\|\mu_x^k\right\|^2} \\
&+ \widehat{\mu_{yy}^k\mu_{xx}^l} + \widehat{\left\|\mu_y^k\right\|^2}\widehat{\left\|\mu_x^l\right\|^2} - \widehat{\mu_{yy}^k}\widehat{\left\|\mu_x^l\right\|^2} - \widehat{\mu_{xx}^l}\widehat{\left\|\mu_y^k\right\|^2},
\end{aligned}
$$

results in a (generally negligible) bias of $\mathcal{O}\left(n^{-1}\right)$ and can be calculated within the time cost $\mathcal{O}\left(n^2\right)$.

### D.2.2 VARIANCE OF $\mathrm{LiNGIC}_u(Z)$ UNDER $\mathcal{H}_0$

The complete proof is given in (Gretton et al., 2007). We show only some of the key steps here. According to (Serfling, 2009, Section 5.2.1), the variance of the U-statistic with the kernel can be calculated by

$$
\mathrm{Var}\left[\mathrm{LiNGIC}_u(\mathcal{D})\right] = \binom{n}{4}^{-1}\sum_{c=1}^{4}\binom{4}{c}\binom{n-4}{4-c}\zeta_c = \frac{4\binom{n-4}{3}}{\binom{n}{4}}\zeta_1 + \frac{6\binom{n-4}{2}}{\binom{n}{4}}\zeta_2 + \mathcal{O}\left(n^{-3}\right),
$$

where we only need to consider the dominant term

$$
\zeta_2 = \mathbb{E}_{i,j}\left[\left(\mathbb{E}_{q,r}h_{ijqr}\right)\right]^2 - \underbrace{\left[\mathbb{E}\,\mathrm{LiNGIC}_u(\mathcal{D})\right]^2}_{0 \text{ under } \mathcal{H}_0}.
$$

using degeneracy $(\zeta_1 = 0)$ under $\mathcal{H}_0$. Under $\mathcal{H}_0$, using $x, y$ are independent, we have

$$
\begin{aligned}
\mathbb{E}_{q,r}h_{ijqr} = &\frac{1}{6}\left(k_{ij}^X + \mathbb{E}_{xx'}k - \mathbb{E}_x k_i - \mathbb{E}_x k_j\right)\left(l_{ij}^Y + \mathbb{E}_{yy'}l - \mathbb{E}_y l_i - \mathbb{E}_y l_j\right) \\
&+ \frac{1}{6}\left(k_{ij}^Y + \mathbb{E}_{yy'}k - \mathbb{E}_y k_i - \mathbb{E}_y k_j\right)\left(l_{ij}^X + \mathbb{E}_{xx'}l - \mathbb{E}_x l_i - \mathbb{E}_x l_j\right).
\end{aligned}
$$

$$(6\mathbb{E}_{q,r}h_{ijqr})^2 =$$

$$
\begin{aligned}
&= \left(k_{ij}^X + \mathbb{E}_{xx'}k - \mathbb{E}_x k_i - \mathbb{E}_x k_j\right)^2 \left(l_{ij}^Y + \mathbb{E}_{yy'}l - \mathbb{E}_y l_i - \mathbb{E}_y l_j\right)^2 \\
&\quad + 2\left(k_{ij}^X + \mathbb{E}_{xx'}k - \mathbb{E}_x k_i - \mathbb{E}_x k_j\right)\left(l_{ij}^X + \mathbb{E}_{xx'}l - \mathbb{E}_x l_i - \mathbb{E}_x l_j\right) \\
&\qquad \cdot \left(l_{ij}^Y + \mathbb{E}_{yy'}l - \mathbb{E}_y l_i - \mathbb{E}_y l_j\right)\left(k_{ij}^Y + \mathbb{E}_{yy'}k - \mathbb{E}_y k_i - \mathbb{E}_y k_j\right) \\
&\quad + \left(k_{ij}^Y + \mathbb{E}_{yy'}k - \mathbb{E}_y k_i - \mathbb{E}_y k_j\right)^2 \left(l_{ij}^X + \mathbb{E}_{xx'}l - \mathbb{E}_x l_i - \mathbb{E}_x l_j\right)^2.
\end{aligned}
$$

$$= \left\|C_{xx}^k\right\|^2 \left\|C_{yy}^l\right\|^2 + 2\left\|C_{xx}^{k,l}\right\|^2 \left\|C_{yy}^{k,l}\right\|^2 + \left\|C_{yy}^k\right\|^2 \left\|C_{xx}^l\right\|^2.$$

where

$$C_{xx}^{k,l} := \mathbb{E}\left[\left(\phi_k(X) - \mu_x^k\right) \otimes \left(\psi_l(X) - \mu_x^l\right)\right], \quad \mu_x^k = \mathbb{E}\phi_k(X), \mu_x^l = \mathbb{E}\psi_l(X).$$

And

$$
\begin{aligned}
\mathbb{E}_{ij}\left(k_{ij}^X + \mathbb{E}_{xx'}k - \mathbb{E}_x k_i - \mathbb{E}_x k_j\right)^2 &= \mathbb{E}_{ij}\left\langle \phi\left(x_i\right) - \mu_x^k, \phi\left(x_j\right) - \mu_x^k\right\rangle^2 \\
&= \mathbb{E}_{ij}\left\langle \left(\phi\left(x_i\right) - \mu_x^k\right) \otimes \left(\phi\left(x_i\right) - \mu_x^k\right), \left(\phi\left(x_j\right) - \mu_x^k\right) \otimes \left(\phi\left(x_j\right) - \mu_x^k\right)\right\rangle_{\mathrm{HS}} := \left\|C_{xx}^k\right\|^2,
\end{aligned}
$$

which, following a similar derivation, we have $\mathbb{E}_{ij}\left(l_{ij}^Y + \mathbb{E}_{yy'}l - \mathbb{E}_y l_i - \mathbb{E}_y l_j\right)^2 = \left\|C_{yy}^l\right\|^2$, $\mathbb{E}_{ij}\left(k_{ij}^Y + \mathbb{E}_{yy'}k - \mathbb{E}_y k_i - \mathbb{E}_y k_j\right)^2 = \left\|C_{yy}^k\right\|^2$, and $\mathbb{E}_{ij}\left(l_{ij}^X + \mathbb{E}_{xx'}l - \mathbb{E}_x l_i - \mathbb{E}_x l_j\right)^2 = \left\|C_{xx}^l\right\|^2$. We also have

$$\mathbb{E}_{ij}\left(k_{ij}^X + \mathbb{E}_{xx'}k - \mathbb{E}_x k_i - \mathbb{E}_x k_j\right)\left(l_{ij}^X + \mathbb{E}_{xx'}l - \mathbb{E}_x l_i - \mathbb{E}_x l_j\right) = \left\|C_{xx}^{k,l}\right\|^2$$

$$\mathbb{E}_{ij}\left(l_{ij}^Y + \mathbb{E}_{yy'}l - \mathbb{E}_y l_i - \mathbb{E}_y l_j\right)\left(k_{ij}^Y + \mathbb{E}_{yy'}k - \mathbb{E}_y k_i - \mathbb{E}_y k_j\right) = \left\|C_{yy}^{k,l}\right\|^2.$$

Then the variance of the statistic is obtained by

$$
\begin{aligned}
\mathrm{Var}\left[\mathrm{LiNGIC}_u(\mathcal{D})\right] = \frac{2(n-4)(n-5)}{(n)_4} &\Big( \left\|C_{xx}^k\right\|_{\mathrm{HS}}^2 \left\|C_{yy}^l\right\|_{\mathrm{HS}}^2 + 2\left\|C_{xx}^{k,l}\right\|_{\mathrm{HS}}^2 \left\|C_{yy}^{k,l}\right\|_{\mathrm{HS}}^2 \\
&\quad + \left\|C_{xx}^l\right\|_{\mathrm{HS}}^2 \left\|C_{yy}^k\right\|_{\mathrm{HS}}^2 \Big) + \mathcal{O}\left(n^{-3}\right),
\end{aligned}
$$

where $\|\cdot\|_{\mathrm{HS}}^2$ is the Hilbert-Schmidt norm. An empirical estimate of the product of Hilbert-Schmidt norms $\left\|C_{xx}^k\right\|_{\mathrm{HS}}^2 \left\|C_{yy}^l\right\|_{\mathrm{HS}}^2$ and $\left\|C_{yy}^k\right\|_{\mathrm{HS}}^2 \left\|C_{xx}^l\right\|_{\mathrm{HS}}^2$ is given by

$$\frac{\mathbf{1}^T(\mathbf{B}_i - \mathrm{diag}(\mathbf{B}_i))\mathbf{1}}{n(n-1)}, \text{ with } \mathbf{B}_1 = ((\mathbf{HK_xH}) \odot (\mathbf{HL_yH}))^{\cdot 2}, \ \mathbf{B}_1 = ((\mathbf{HK_yH}) \odot (\mathbf{HL_xH}))^{\cdot 2},$$

respectively, where $\odot$ is the entrywise matrix product and $()^{\cdot 2}$ is the entrywise matrix power. For $\left\|C_{xx}^{k,l}\right\|_{\mathrm{HS}}^2 \left\|C_{yy}^{k,l}\right\|_{\mathrm{HS}}^2$, we first give the unbiased estimators (where $\tilde{K} = HKH$ and $\tilde{L} = HLH$):

$$\widehat{\left\|C_{xx}^{k,l}\right\|}_{\mathrm{HS}}^2 = \frac{1}{n(n-1)}\sum_{i \neq j} \tilde{K}_{ij}^X \tilde{L}_{ij}^X = \frac{1}{n(n-1)}\mathbf{1}^\top((\tilde{K}^X \odot \tilde{L}^X) - \mathrm{diag}(\tilde{K}^X \odot \tilde{L}^X))\mathbf{1}$$

$$\widehat{\left\|C_{yy}^{k,l}\right\|}_{\mathrm{HS}}^2 = \frac{1}{n(n-1)}\sum_{i \neq j} \tilde{K}_{ij}^Y \tilde{L}_{ij}^Y = \frac{1}{n(n-1)}\mathbf{1}^\top((\tilde{K}^Y \odot \tilde{L}^Y) - \mathrm{diag}(\tilde{K}^Y \odot \tilde{L}^Y))\mathbf{1}.$$

Therefore the empirical estimate of $\left\|C_{xx}^{k,l}\right\|_{\mathrm{HS}}^2 \left\|C_{yy}^{k,l}\right\|_{\mathrm{HS}}^2$ is given by the formula above with

$$\mathbf{B}_3 = (\mathbf{HK_XH}) \odot (\mathbf{HL_XH}) \odot (\mathbf{HK_YH}) \odot (\mathbf{HL_YH}).$$

The estimate in has a bias of $\mathcal{O}\left(n^{-3}\right)$ and can be calculated within time cost $\mathcal{O}\left(n^2\right)$. Since the additional terms of the bias vanish faster than the terms in front of it, the result is identical to the case of unbiased.

# E    SUPPLEMENTARY EXPERIMENTAL DETAILS AND RESULTS

## E.1    IMPLEMENTATION DETAILS

Here we provide the implementation details of the methods. In all experiments, we use Gaussian kernels in all kernel-based methods. The significance level is set to 0.05. The results are obtained after averaging the values in the 100 tests.

**Details about the Baselines.** All the baselines follow their default settings unless stated otherwise. **HSIC** (Gretton et al., 2007): the original HSIC test using gamma approximation for $p$-value. Code from python library causal-learn (Zheng et al., 2024); **LFHSIC** (Ren et al., 2024): HSIC test with adaptively learned bandwidth. Code from https://github.com/renyixin666/HSIC-LK; **RDC** (Lopez-Paz et al., 2013): use canonical correlation between a finite set of random Fourier features. We permute the samples 500 times to compute the empirical $p$-value. Code from https://github.com/garydoranjr/rdc; **FHSIC** (Zhang et al., 2018): HSIC using finite-dimensional random Fourier feature mappings to approximate kernels. Code from https://github.com/oxcsml/kerpy.

## E.2    MORE RESULTS ON SYNTHETIC DATA

### E.2.1    EXPERIMENT RESULTS ON MORE BASELINES

We also conducted the same synthetic experiments on two more baselines, RDC (Lopez-Paz et al., 2013) and MI (Berrett & Samworth, 2019). The results are shown in the Figure 6, 5, 8, and 7.

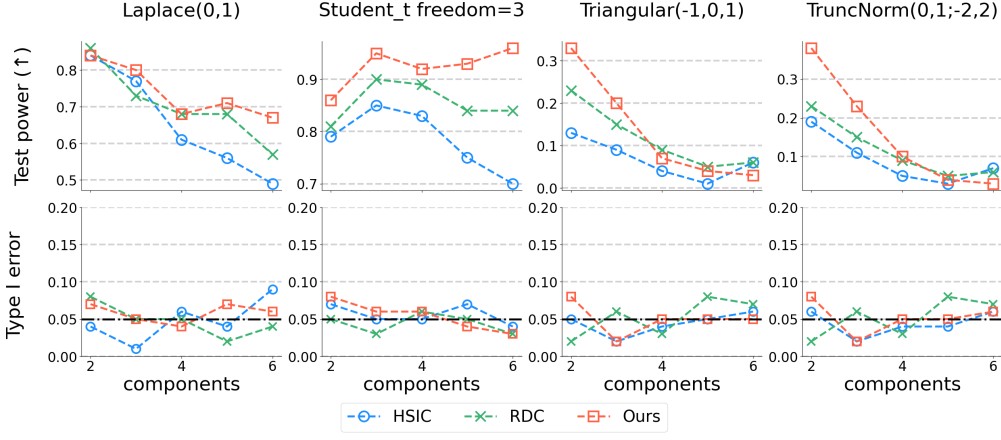

Figure 5: The experiment results when we change the number of the independent components of the linear mixtures with 500 samples. The number of components $d \in \{2, 3, 4, 5, 6\}$. Each column shows the results with $\varepsilon_i \sim$ a different distribution. The first row demonstrates Test Power and the second row shows the Type I error. The significance level 0.05 is annotated as the black line.

SCIT is designed for conditional independence testing under linear non-Gaussian SEMs by regressing out conditioning variables and applying a generic nonparametric kernel-based independence check on residuals. When setting $Z = \emptyset$, SCIT reduces to testing whether the coordinate-wise RBF similarity between $(x, y)$ is significantly larger than that between $(x, r)$, where $r$ is a permuted copy of $y$. This generic independence test does not leverage the linear non-Gaussian structure as our method does. We added SCIT with $Z = \emptyset$ as one of our baselines in the Table 3.

### E.2.2    EXPERIMENT RESULTS ON PERMUTATION TESTING

To show the performance of our derived asymptotic distribution, we also run experiments on HSIC and LiNGIC with a permutation test. The number of permutations equals to 500. The results are shown in the Figure 9 and 10. Here we also included HSIC_chi as one of the baselines, which means the null distribution approximated by the sum of chi-square variables as mentioned in (Zhang et al., 2012).

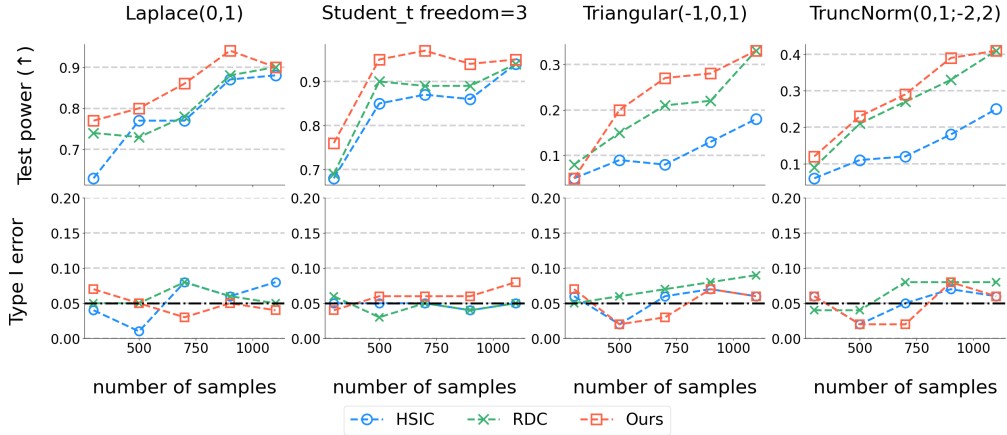

Figure 6: The experiment results when we change the sample sizes of the linear mixtures of 3 independent components from different distributions. The sample sizes $n \in \{300, 500, 700, 900, 1100\}$.

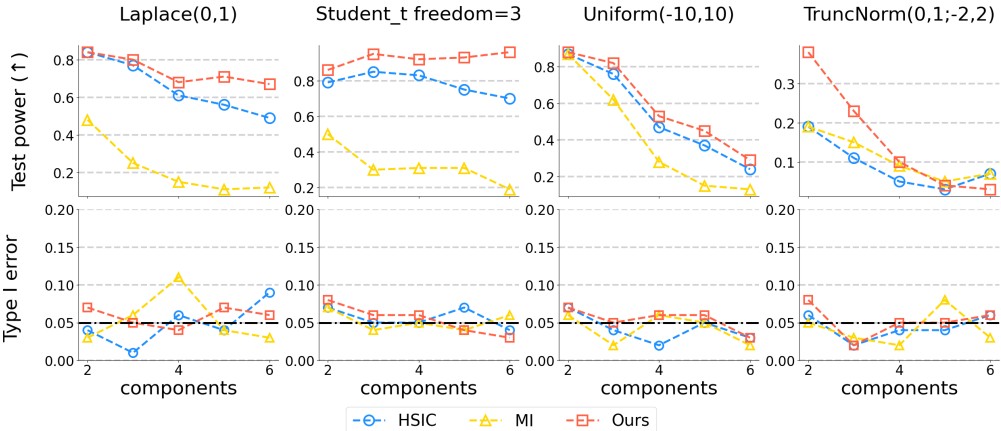

Figure 7: The experiment results when we change the number of the independent components of the linear mixtures with $500$ samples. The number of components $d \in \{2, 3, 4, 5, 6\}$. Each column shows the results with $\varepsilon_i \sim$ a different distribution. The first row demonstrates Test Power and the second row shows the Type I error. The significance level $0.05$ is annotated as the black line.

### E.2.3 SENSITIVITY TO DEPENDENCE STRENGTH

We conducted additional experiments where all parameters are fixed and only the dependence strength between $X$ and $Y$ is varied. Specifically, our data generation process is defined as:

$$Y = c \cdot S_Y + \sqrt{1 - c^2} \cdot N_Y, \quad X = c \cdot S_X + \sqrt{1 - c^2} \cdot N_X, \tag{24}$$

where $S_X, S_Y$ are generated from shared latent sources, and $N_X, N_Y$ are generated from independent sources. Thus, $c = 0$ corresponds to independence ($X \perp Y$), while $c = 1$ yields the strongest dependence (entirely shared source).

We report results (with $d = 3$ independent components drawn from a Student-$t$ distribution and sample size $n = 500$) in Table 4.

### E.3 COMPUTATIONAL COMPLEXITY AND RUNTIME ANALYSIS

The computational cost of our independence test, LiNGIC, is comparable to standard kernel-based independence tests such as HSIC, as it primarily involves estimating the Hilbert-Schmidt norm of a covariance operator in the Reproducing Kernel Hilbert Space (RKHS).

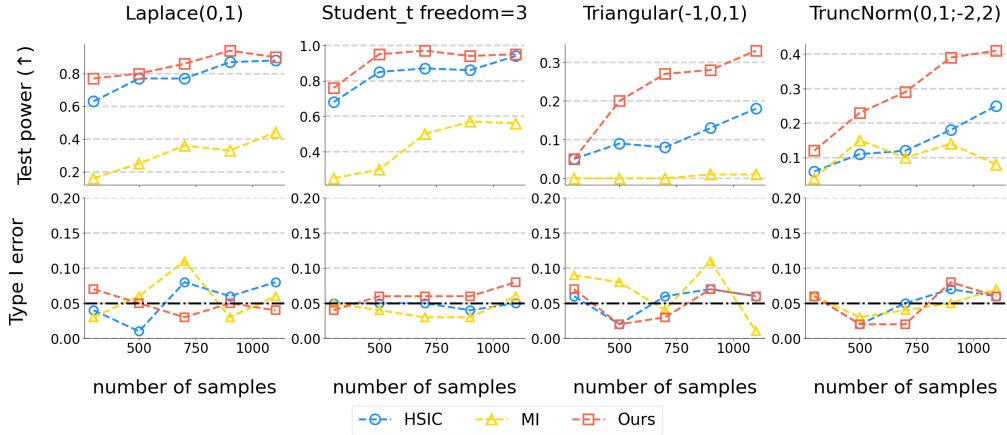

Figure 8: The experiment results when we change the sample sizes of the linear mixtures of 3 independent components from different distributions. The sample sizes $n \in \{300, 500, 700, 900, 1100\}$.

Table 3: Comparison of Type I error and Power across different distributions and dimensions ($d$).

| Distribution | $d$ | HSIC | | SCIT | | LiNGIC | |
|---|---|---|---|---|---|---|---|
| | | Type I | Power | Type I | Power | Type I | Power |
| Laplace | 2 | 0.04 | 0.84 | 0.07 | 0.57 | 0.07 | 0.84 |
| | 3 | 0.01 | 0.77 | 0.01 | 0.61 | 0.04 | 0.80 |
| | 4 | 0.06 | 0.61 | 0.01 | 0.60 | 0.05 | 0.67 |
| | 5 | 0.04 | 0.56 | 0.06 | 0.51 | 0.07 | 0.70 |
| | 6 | 0.09 | 0.49 | 0.05 | 0.39 | 0.06 | 0.66 |
| Student t | 2 | 0.07 | 0.79 | 0.02 | 0.54 | 0.08 | 0.85 |
| | 3 | 0.05 | 0.85 | 0.06 | 0.69 | 0.06 | 0.97 |
| | 4 | 0.05 | 0.83 | 0.03 | 0.62 | 0.06 | 0.95 |
| | 5 | 0.07 | 0.75 | 0.06 | 0.60 | 0.04 | 0.93 |
| | 6 | 0.04 | 0.70 | 0.07 | 0.57 | 0.03 | 0.96 |
| Uniform | 2 | 0.07 | 0.87 | 0.05 | 0.00 | 0.07 | 0.88 |
| | 3 | 0.04 | 0.76 | 0.07 | 0.01 | 0.05 | 0.83 |
| | 4 | 0.02 | 0.47 | 0.06 | 0.00 | 0.06 | 0.54 |
| | 5 | 0.05 | 0.37 | 0.03 | 0.00 | 0.06 | 0.45 |
| | 6 | 0.03 | 0.24 | 0.04 | 0.00 | 0.03 | 0.31 |
| Truncnorm | 2 | 0.06 | 0.19 | 0.12 | 0.01 | 0.08 | 0.41 |
| | 3 | 0.02 | 0.11 | 0.01 | 0.01 | 0.02 | 0.25 |
| | 4 | 0.04 | 0.05 | 0.06 | 0.01 | 0.05 | 0.12 |
| | 5 | 0.04 | 0.03 | 0.04 | 0.00 | 0.05 | 0.04 |
| | 6 | 0.06 | 0.07 | 0.05 | 0.00 | 0.06 | 0.03 |

Specifically, given $n$ samples, computing the $n \times n$ Gram matrices requires $\mathcal{O}(n^2)$ kernel evaluations. As discussed in Gretton et al. (2005a), the centering and evaluation of the V-statistic $\mathrm{Tr}(\mathbf{K}_X \mathbf{H} \mathbf{L}_Y \mathbf{H})$ can be implemented using three double sums, each of which is $\mathcal{O}(n^2)$. Consequently, a single LiNGIC evaluation has an overall time complexity of $\mathcal{O}(n^2)$ (quadratic in sample size), matching the complexity of standard kernel-based independence tests.

Importantly, our main contribution is the improved statistical power under the linear non-Gaussian setting rather than computational acceleration. Nevertheless, LiNGIC operates on a comparable runtime scale to the widely used HSIC. Designing a faster variant of LiNGIC (e.g., via random Fourier features or Nyström approximation) remains a potential direction for future work.

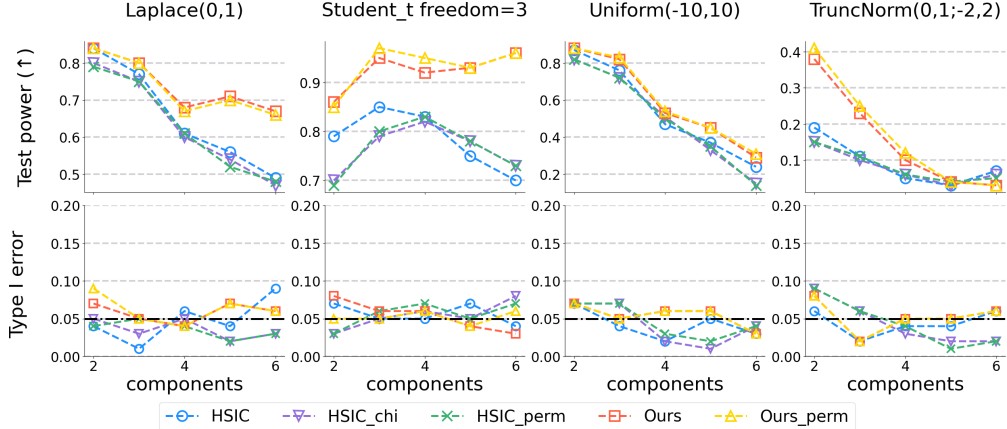

Figure 9: The experiment results when we change the number of the independent components of the linear mixtures with 500 samples. The number of components $d \in \{2, 3, 4, 5, 6\}$. Each column shows the results with $\varepsilon_i \sim$ a different distribution. The first row demonstrates Test Power and the second row shows the Type I error. The significance level $0.05$ is annotated as the black line.

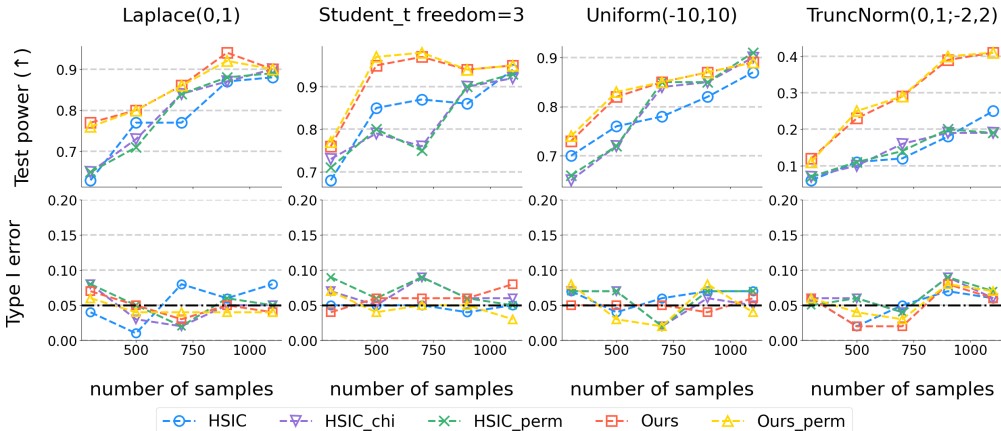

Figure 10: The experiment results when we change the sample sizes of the linear mixtures of 3 independent components from different distributions. The sample sizes $n \in \{300, 500, 700, 900, 1100\}$.

For empirical evaluation, we compared the runtime (in seconds) of our testing method against baselines, as well as their corresponding applications in causal discovery (specifically using Direct-LiNGAM). The results are detailed below.

As shown in Table 5 and Table 6, we observe that LiNGIC operates on a similar timescale as classical kernel-based tests.

### E.4 SACHS DATASET AND RESULTS OF THE REAL DATA

**Sachs Dataset.** The the flow-cytometry data published by (Sachs et al., 2005) is a popular real-world data set for causal discovery methods, which gives expression levels of proteins under various experimental conditions. Among all data, we use $n = 853$ observational measurements of 11 proteins. We take the popular reconstruction of the signaling network in original SAHCS dataset paper (the right graph in Fig. 11) as the ground-truth causal graph, which contains 11 nodes and 17 directed edges.

Table 4: Power comparison (higher is better) under varying dependence strength $c$. The last column reports the Type I error at $c = 0$ (lower is better, target $\alpha = 0.05$).

| Method | Dependence Strength ($c$) | | | | | | | | | | Type I |
| --- | --- | --- | --- | --- | --- | --- | --- | --- | --- | --- | --- |
| | 0.1 | 0.2 | 0.3 | 0.4 | 0.5 | 0.6 | 0.7 | 0.8 | 0.9 | 1.0 | ($c = 0$) |
| HSIC | 0.01 | 0.04 | 0.05 | 0.10 | 0.17 | 0.28 | 0.37 | 0.53 | 0.72 | 0.83 | 0.05 |
| LiNGIC | **0.08** | **0.06** | **0.12** | **0.23** | **0.39** | **0.53** | **0.69** | **0.86** | **0.94** | **0.95** | 0.05 |

Table 5: Runtime comparison (in seconds) of different independence tests with varying sample sizes ($n$). "Perm" denotes permutation-based tests (shuffles=100).

| Sample Size ($n$) | HSIC | HSIC$_{\text{perm}}$ | LiNGIC | LiNGIC$_{\text{perm}}$ | RDC |
| --- | --- | --- | --- | --- | --- |
| 500 | 0.011 | 2.693 | 0.036 | 7.159 | 2.190 |
| 1000 | 0.051 | 16.267 | 0.150 | 36.427 | 2.478 |
| 2000 | 0.205 | 66.178 | 0.499 | 102.564 | 3.131 |
| 3000 | 0.411 | 166.481 | 1.154 | 233.445 | 3.778 |
| 4000 | 0.678 | 301.443 | 2.016 | 405.439 | 4.352 |
| 5000 | 0.995 | 483.046 | 3.126 | 638.219 | 4.773 |

### E.5 REAL DATA EXPERIMENT RESULTS

### E.6 MORE RESULTS ON DIRECT-LiNGAM

### E.7 ADDITIONAL SIMULATION SETTINGS AND RESULTS

We conducted new simulations similar to the settings described in Section 7.1 of Shimizu et al. (2006). We evaluated our method against the baseline (DirectLiNGAM with HSIC) under varying graph structures (sparse vs. dense), dimensionalities, and non-Gaussian noise distributions.

**Data Generation.** We simulated both **Sparse (S)** graphs (with an average degree of 2) and **Dense (D)** graphs (with an edge probability $p = 0.7$). We also varied the dimension of the dataset (number of nodes) within $d \in \{5, 10, 15\}$. We used strictly lower-triangular matrices $\mathbf{B}$ with weights sampled uniformly from $[-1.5, -0.5] \cup [0.5, 1.5]$. The disturbance terms were generated from **non-Gaussian distributions** (e.g., Uniform, Student-$t$, and Truncnorm), followed by a random permutation of the causal order.

**Experimental Results.** We report the performance mean over 100 trials on datasets with $N = 200$ samples. Our method demonstrates strong generalizability and consistently maintains higher performance with respect to the F1 score ($\uparrow$) and SHD ($\downarrow$).

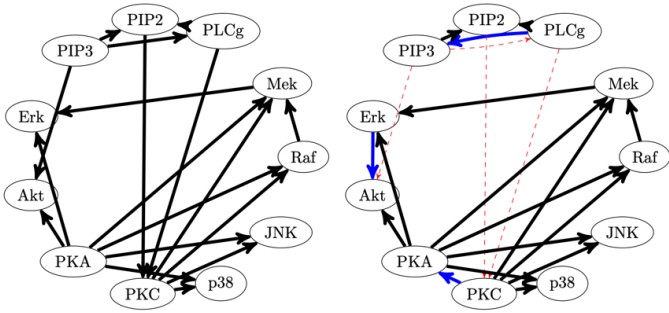

Figure 11: The first two figures in Figure 5 in (Mooij & Heskes, 2013). **Left:** Consensus network, according to (Sachs et al., 2005); **Right:** Reconstruction network, reconstruction of the signaling network by (Sachs et al., 2005), in comparison with the consensus network. Black edges: expected. Blue edges: unexpected, novel findings. Red dashed edges: missing.

Table 6: Runtime comparison (in seconds) of Direct-LiNGAM using HSIC vs. LiNGIC as the independence measure ($N = 200$).

| Settings | Direct-LiNGAM (HSIC) | Direct-LiNGAM (LiNGIC) |
|---|---|---|
| $d = 20$ | 4.75 | 7.05 |
| $d = 50$ | 62.46 | 99.52 |
| $d = 100$ | 500.17 | 802.77 |

Table 7: SHD and F1 Score of Direct-LiNGAM algorithm using different testing methods on the ground-truth data of the SACHS dataset.

| Sachs | MI | HSIC | RDC | HSIC-RFF | dCor | LiNGIC |
|---|---|---|---|---|---|---|
| F1 ($\uparrow$) | 0.18 | 0.10 | 0.18 | 0.10 | 0.29 | 0.22 |
| SHD ($\downarrow$) | 15 | 16 | 15 | 16 | 14 | 15 |

### E.8 PERFORMANCE ON HIGH-DIMENSIONAL GRAPHS

We conducted additional simulations to evaluate our method on high-dimensional causal graphs with the number of nodes $d \in \{20, 50, 100\}$.

**Data Generation.** We simulated graphs with an average degree of 2. We used strictly lower-triangular matrices $\mathbf{B}$ with weights sampled uniformly from $[-1.5, -0.5] \cup [0.5, 1.5]$. The disturbance terms were generated from Uniform or Truncnorm distributions, followed by a random permutation of the causal order.

**Experimental Results.** We report the performance mean over 100 trials on datasets with $n = 200$ samples. Our method demonstrates strong generalizability and consistently maintains higher performance with respect to the F1 score ($\uparrow$) and SHD ($\downarrow$).

## F DISCUSSIONS

### F.1 THE PREVALANCE OF NON-GAUSSIAN DISTRIBTUTIONS

In fact, even we do not care about the non-Gaussianity requirement in causal discovery, non-Gaussian data are far more prevalent than Gaussian ones in the real world, as mentioned in (Spirtes & Zhang, 2016). According to the Cramér's decomposition theorem (Cramér, 1970), if any of the variables with non-zero coefficient in the linear composition is non-Gaussian, then the composition must be non-Gaussian. This implies the rareness of the Gaussian distribution in linear data since it is easily "polluted" by non-Gaussian ones.

### F.2 JUSTIFICATION FOR PAIRWISE INDEPENDENCE TESTING

In our linear non-Gaussian setting, joint independence can indeed be reduced to pairwise independence. This relies on a fundamental result in Independent Component Analysis (ICA). As stated in Comon (1994):

**Proposition F.1.** *Let $\mathbf{s}$ be a random vector with mutually independent components, and let $\mathbf{x} = \mathbf{G}\mathbf{s}$. Then, the mutual independence of the entries of $\mathbf{x}$ is equivalent to their pairwise independence.*

In our framework, the variables are linear mixtures of independent noise terms. Specifically, we can write:

$$Y = \sum_i a_i \varepsilon_i = \mathbf{a}^\top \boldsymbol{\varepsilon}, \quad X = \sum_i b_i \varepsilon_i = \mathbf{b}^\top \boldsymbol{\varepsilon}, \quad Z = \sum_i c_i \varepsilon_i = \mathbf{c}^\top \boldsymbol{\varepsilon}, \tag{25}$$

Table 8: Comparison of F1 Score (higher is better) and SHD (lower is better) on different settings.

| Settings | Nodes (d) | F1 Score (↑) | | SHD (↓) | |
|---|---|---|---|---|---|
| | | HSIC | Ours | HSIC | Ours |
| Student t (D) | 5 | 0.906 | **0.909** | 0.93 | **0.86** |
| | 10 | 0.717 | **0.754** | 13.11 | **11.43** |
| | 15 | 0.614 | **0.652** | 43.3 | **39.21** |
| Student t (S) | 5 | 0.913 | **0.935** | 0.58 | **0.43** |
| | 10 | 0.833 | **0.847** | 2.82 | **2.52** |
| | 15 | 0.756 | **0.773** | 7.36 | **6.53** |
| Uniform (D) | 5 | 0.905 | **0.927** | 0.92 | **0.72** |
| | 10 | 0.620 | **0.700** | 17.91 | **14.21** |
| | 15 | 0.508 | **0.566** | 53.95 | **48.21** |
| Uniform (S) | 5 | 0.945 | **0.968** | 0.49 | **0.36** |
| | 10 | 0.916 | **0.927** | 1.72 | **1.51** |
| | 15 | 0.841 | **0.875** | 4.52 | **3.49** |
| Truncnorm (D) | 5 | 0.474 | **0.572** | 4.78 | **3.95** |
| | 10 | **0.435** | 0.434 | 25.92 | **25.75** |
| | 15 | 0.412 | **0.450** | 63.31 | **59.84** |
| Truncnorm (S) | 5 | 0.506 | **0.613** | 3.52 | **2.81** |
| | 10 | 0.442 | **0.556** | 10.84 | **8.98** |
| | 15 | 0.402 | **0.432** | 19.18 | **18.93** |

Table 9: Comparison of F1 Score and SHD on high-dimensional graphs ($d \in \{20, 50, 100\}$).

| Distribution | Nodes (d) | F1 Score (↑) | | SHD (↓) | |
|---|---|---|---|---|---|
| | | HSIC | Ours | HSIC | Ours |
| Uniform | 20 | 0.828 | **0.835** | 6.9 | **6.5** |
| | 50 | 0.550 | **0.613** | 58.9 | **51.1** |
| | 100 | 0.363 | **0.454** | 191.0 | **159.0** |
| Truncnorm | 20 | 0.422 | **0.489** | 24.3 | **22.3** |
| | 50 | 0.296 | **0.367** | 119.3 | **101.9** |
| | 100 | 0.283 | **0.358** | 249.7 | **208.6** |

where $\mathbf{a}, \mathbf{b}, \mathbf{c}$ are coefficient vectors and $\varepsilon$ is a vector of mutually independent non-Gaussian components. This system can be expressed in matrix form as:

$$\begin{pmatrix} Y \\ X \\ Z \end{pmatrix} = \begin{bmatrix} \mathbf{a}^\top \\ \mathbf{b}^\top \\ \mathbf{c}^\top \end{bmatrix} \varepsilon \triangleq \mathbf{G}\varepsilon. \tag{26}$$

Since $\varepsilon$ has mutually independent components, the condition of the proposition is satisfied. Hence, the mutual (joint) independence of $(Y, X, Z)$ is equivalent to their pairwise independence under our model.

Therefore, to test the joint independence of $(Y, X, Z)$, it is sufficient to apply LiNGIC to the pairs $(Y, X)$, $(Y, Z)$, and $(X, Z)$ and combine the results accordingly. Extending LiNGIC to a direct multivariate test in this setting (in the spirit of dHSIC) remains an interesting direction for future work.

### F.3 LIMITATIONS

We acknowledge that our theoretical guaranties rely on the assumption of linear mixtures of non-Gaussian variables. Extending the setting to more general nonlinear mixing models would be a non-trivial task and represents a promising avenue for future research.

