# OpenReview forum: "Independence Test for Linear Non-Gaussian Data and Applications in Causal Discovery"
_ICLR.cc/2026/Conference — ICLR 2026 Poster_

### Official Review · Reviewer_GaAN · 2025-10-28

**Soundness:** 2
**Presentation:** 3
**Contribution:** 3
**Rating:** 4
**Confidence:** 4

**Summary:**

This paper investigates the problem of independence testing under linear non-Gaussian data, which is an important problem in both statistics and machine learning. Under the framework where variables exhibit linear relationships and the noise terms follow non-Gaussian distributions, the authors provide a new theoretical characterization of independence, proving that the constancy of conditional mean and conditional variance is sufficient to guarantee independence in this specific model. Based on this insight, they develop a test statistic and derive its asymptotic distribution. Finally, extensive experiments on both synthetic and real-world datasets are conducted to validate the effectiveness of the proposed method.

**Strengths:**

Originality: This paper provides a new theoretical characterization of the independence testing problem under the framework of linear non-Gaussian models. It introduces an innovative perspective by proposing that the constancy of conditional mean and conditional variance serves as a sufficient condition for independence, which represents a clear conceptual advancement.

Quality: The theoretical derivations in the paper are rigorous and logically coherent. The authors not only define the test statistic and analyze its consistency but also derive its asymptotic distribution, thereby providing a solid theoretical foundation for significance evaluation.

Clarity: The paper is clearly written and well-structured, presenting a coherent flow from problem motivation and theoretical formulation to methodological development and experimental validation. The notation is well-defined, and the main theorems and lemmas are easy to follow.

Significance: The proposed work offers a new theoretical tool for independence inference under non-Gaussian conditions, providing a strong complement to existing independence testing methods that rely on Gaussian assumptions.

**Weaknesses:**

1. The paper lacks an empirical comparison with RDC [1] and MI [2], two methods capable of detecting a wide range of dependency types.
2. Some relevant literature on independence testing and dependence measurement is missing, such as references [3] and [4].
3. The paper claims to have conducted experiments on the SACHS real-world dataset, but the corresponding results are not reported. In addition, three figures are provided in the Appendix E.3, yet it is unclear which one is used as the ground truth. If only the first figure in Figure 7 is used, the other two seem unrelated to the paper.
4. There are no empirical results for cases where $X$ and $Y$ are high-dimensional. To my knowledge, the main challenges in independence testing arise from nonlinearity and high dimensionality, but the paper presents no results concerning the high-dimensional setting.
5. The definitions of the two evaluation metrics in Table 1 are not formally provided.

Refs: [1] The randomized dependence coefficient. Advances in neural information processing systems, 2013.
[2] Nonparametric independence testing via mutual information. Biometrika, 2019.
[3] Distribution-free tests of independence in high dimensions. 2017, Biometrika.
[4] A new coefficient of correlation. 2021, Journal of the American Statistical Association.

**Questions:**

1. As stated in line 336 of the paper, I am very interested in seeing how the proposed method performs when using a permutation approach. Specifically, how do the Type I error rate, testing power, and computational efficiency compare with the current approach that uses a Gamma distribution to approximate the null distribution?
2. When evaluating the test power in independence testing, could you fix the parameters $a_i$ and $b_i$, and examine how the testing power of the proposed method changes as they vary from small values (e.g., 0.1) to large values (e.g., 1)?
3. Could you provide more simulation experiments when applying the proposed method to causal discovery, similar to the experimental setups presented in the “Simulations” section of reference [1]? Using only the causal graph corresponding to real-world data does not demonstrate the generalizability of the proposed method to different graph structures.
4. Line 278 of the paper states that both Gaussian and polynomial kernels are used, but the simulations emphasize that only the Gaussian kernel is adopted, which is quite confusing.

Ref: [1] A linear non-Gaussian acyclic model for causal discovery. Journal of Machine Learning Research, 2006.

---

> ### Author Response · Authors · 2025-11-25
> **Response to Reviewer GaAN (Part 1/4)**
>
> We sincerely appreciate the reviewer's constructive comments and helpful feedback. Please see below for our response.
>
> ---
>
> **(Q1)** The reviewer asks for clarification on several experimental details:
>
> - **(Q1.1)**  The definitions of the two evaluation metrics in Table 1.
>
>   **A:** Thank you for raising this. We added the following discussions in our manuscript (in Section 6.2):
>
>   > We report the Structural Hamming Distance (SHD) and F1 score. SHD measures how many edge insertions, deletions, or reversals are needed to transform the estimated graph into the ground-truth graph. The F1 score is the harmonic mean of precision and recall, summarizing how accurately and completely the estimated edges recover the true causal edges.
>
> - **(Q1.2)** The experimental results on the SACHS dataset and the ground truth graph.
>
>   **A:** Thank you for pointing this out. The experimental results on the SACHS dataset are **included in Appendix E.4** (Table 3 in the updated manuscript), and we have now made this reference explicit in the main text.
>
>   Regarding the ground-truth graph used for evaluation, we have clarified in the revision that the **reconstruction network** (the right graph in Fig.7, containing 11 nodes and 17 directed edges) is the one used for quantitative evaluation. To avoid confusion, we removed the previously redundant figure and kept only two: (1) the consensus network from the original paper [[S+05\]](https://www.science.org/doi/10.1126/science.1105809), which is included to provide readers with additional context about the dataset, and (2) the reconstruction network used for evaluation.
>
> - **(Q1.3)** Confusion about Gaussian vs. polynomial kernels.
>
>   **A:** Thank you for your careful reading. To clarify: for $\operatorname{LiNGIC}_1(X, Y)$, it **indeed uses a Gaussian kernel for $X$ and a second-order polynomial kernel for $Y$**, as this choice follows directly from Theorem 4.3, where the statistic is based on the covariance between $f(X)$ and $(Y, Y^2)$. Thus, the polynomial kernel is not an arbitrary option but an inherent part of our design.
>
>   The "Gaussian kernel" mentioned in the simulation section (line 393~394) refers **only to the baselines that require characteristic kernels** (HSIC, HSIC-RFF, LFHSIC). We have rewritten the relevant sentence to avoid this confusion.
>
> - **(Q1.4)** Empirical comparison with RDC [[L+13\]](https://arxiv.org/pdf/1304.7717) and MI [[B+19\]](https://arxiv.org/pdf/1711.06642).
>
>   **A:** Thank you for raising this. We have now included RDC and MI as two of our baselines in Appendix E.2, Figure 5~8. Below we list the comparison of Student t distribution between LiNGIC, RDC, and MI:
>
>   | Method            | LiNGIC     |           | RDC        |           | MI         |           |
>   | :---------------- | :--------- | :-------- | :--------- | :-------- | :--------- | :-------- |
>   | #components       | **Type I($\downarrow$)** | **Power($\uparrow$)** | **Type I($\downarrow$)** | **Power($\uparrow$)** | **Type I($\downarrow$)** | **Power($\uparrow$)** |
>   | Student t (d=2)   | 0.08       | **0.85**      | 0.05       | 0.81      | 0.07       | 0.5       |
>   | (d=3)             | 0.06       | **0.97**      | 0.03       | 0.9       | 0.04       | 0.3       |
>   | (d=4)             | 0.06       | **0.95**      | 0.06       | 0.89      | 0.05       | 0.31      |
>   | (d=5)             | 0.04       | **0.93**      | 0.05       | 0.84      | 0.04       | 0.31      |
>   | (d=6)             | 0.03       | **0.96**      | 0.03       | 0.84      | 0.06       | 0.19      |
>   | Student t (n=300) | 0.08       | **0.77**      | 0.06       | 0.69      | 0.05       | 0.25      |
>   | (n=500)           | 0.06       | **0.97**      | 0.03       | 0.9       | 0.04       | 0.3       |
>   | (n=700)           | 0.06       | **0.98**      | 0.05       | 0.89      | 0.03       | 0.5       |
>   | (n=900)           | 0.04       | **0.94**      | 0.04       | 0.89      | 0.03       | 0.57      |
>   | (n=1100)          | 0.03       | **0.95**      | 0.05       | 0.94      | 0.06       | 0.56      |

---

> ### Author Response · Authors · 2025-11-25
> **Response to Reviewer GaAN (Part 2/4)**
>
> **(Q2)** The reviewer suggests several additional experiments.
>
> - **(Q2.1)** Type I error, testing power, and computational efficiency compared to a permutation approach.
>
>   **A:** Thank you for raising this. We have added permutation-calibrated results for both LiNGIC and HSIC, and summarize the results below (corresponding figure and the results of varying sample sizes can be found in Appendix E.2). These experiments show that the permutation-based method yields almost the same test power and Type I error as the Gamma approximation, and both versions of LiNGIC consistently achieve the highest power among all baselines.
>
>   |     Method      |    HSIC    |           | HSIC_perm  |           |   LiNGIC   |           | LiNGIC_perm |           |
>   | :-------------: | :--------: | :-------: | :--------: | :-------: | :--------: | :-------: | :---------: | :-------: |
>   |   #components   | **Type I** | **Power** | **Type I** | **Power** | **Type I** | **Power** | **Type I**  | **Power** |
>   |  Laplace (d=2)  |    0.04    |   0.84    |    0.04    |   0.79    |    0.07    |   0.84    |    0.04     |   0.84    |
>   |      (d=3)      |    0.01    |   0.77    |    0.05    |   0.75    |    0.04    |    0.8    |    0.05     |    0.8    |
>   |      (d=4)      |    0.06    |   0.61    |    0.04    |   0.61    |    0.05    |   0.67    |    0.04     |   0.68    |
>   |      (d=5)      |    0.04    |   0.56    |    0.02    |   0.52    |    0.07    |    0.7    |    0.07     |   0.71    |
>   |      (d=6)      |    0.09    |   0.49    |    0.03    |   0.48    |    0.06    |   0.66    |    0.06     |   0.67    |
>   | Student t (d=2) |    0.07    |   0.79    |    0.03    |   0.69    |    0.08    |   0.85    |    0.05     |   0.86    |
>   |      (d=3)      |    0.05    |   0.85    |    0.06    |    0.8    |    0.06    |   0.97    |    0.05     |   0.95    |
>   |      (d=4)      |    0.05    |   0.83    |    0.07    |   0.83    |    0.06    |   0.95    |    0.06     |   0.92    |
>   |      (d=5)      |    0.07    |   0.75    |    0.05    |   0.78    |    0.04    |   0.93    |    0.04     |   0.93    |
>   |      (d=6)      |    0.04    |    0.7    |    0.07    |   0.73    |    0.03    |   0.96    |    0.06     |   0.96    |
>   |  Uniform (d=2)  |    0.07    |   0.87    |    0.07    |   0.82    |    0.07    |   0.88    |    0.07     |   0.88    |
>   |      (d=3)      |    0.04    |   0.76    |    0.07    |   0.72    |    0.05    |   0.83    |    0.05     |   0.82    |
>   |      (d=4)      |    0.02    |   0.47    |    0.03    |    0.5    |    0.06    |   0.54    |    0.05     |   0.53    |
>   |      (d=5)      |    0.05    |   0.37    |    0.02    |   0.35    |    0.06    |   0.45    |    0.06     |   0.45    |
>   |      (d=6)      |    0.03    |   0.24    |    0.04    |   0.14    |    0.03    |   0.31    |    0.05     |   0.29    |
>   | Truncnorm (d=2) |    0.06    |   0.19    |    0.09    |   0.15    |    0.08    |   0.41    |    0.06     |   0.38    |
>   |      (d=3)      |    0.02    |   0.11    |    0.06    |   0.11    |    0.02    |   0.25    |    0.03     |   0.23    |
>   |      (d=4)      |    0.04    |   0.05    |    0.04    |   0.06    |    0.05    |   0.12    |    0.06     |    0.1    |
>   |      (d=5)      |    0.04    |   0.03    |    0.01    |   0.04    |    0.05    |   0.04    |    0.06     |   0.04    |
>   |      (d=6)      |    0.06    |   0.07    |    0.02    |   0.05    |    0.06    |   0.03    |    0.06     |   0.03    |
>
>   We also compare the empirical computational cost of LiNGIC and LiNGIC_perm, as shown in the table below (the average runtime in seconds).  Since permutation-based procedures have substantially higher computational cost while providing no noticeable advantage over the Gamma approximation, we use the latter for efficiency.
>
>   | Sample Size \ Method | HSIC  | HSIC_perm | LiNGIC | LiNGIC_perm |
>   | :------------------: | :---- | --------- | ------ | ----------- |
>   |        n=500         | 0.011 | 2.693     | 0.036  | 7.159       |
>   |        n=1000        | 0.051 | 16.267    | 0.150  | 36.427      |
>   |        n=2000        | 0.205 | 66.178    | 0.499  | 102.564     |
>   |        n=3000        | 0.411 | 166.48    | 1.154  | 233.445     |
>   |        n=4000        | 0.678 | 301.433   | 2.016  | 405.439     |
>   |        n=5000        | 0.995 | 483.046   | 3.126  | 638.219     |

---

> ### Author Response · Authors · 2025-11-25
> **Response to Reviewer GaAN (Part 3/4)**
>
> - **(Q2.2)** could you fix the parameters, and examine how the testing power of the proposed method changes as they vary from small values (e.g., 0.1) to large values (e.g., 1)?
>
>   **A:** Thank you for the constructive suggestion. Following your advice, we added experiments where all parameters are fixed and only the dependence strength between $X$ and $Y$ is varied. Specifically, our data generation process now becomes $Y=c\cdot S_Y + \sqrt{1-c^2}\cdot N_Y, X=c\cdot S_X + \sqrt{1-c^2}\cdot N_X$, where $S_X,S_Y$ are generated from shared latent sources and $N_X,N_Y$ from independent sources. Thus, $c=0$ corresponds to independence $(X\perp Y)$, while $c=1$ yields the strongest dependence (entirely shared source). Below we report part of the results (Testing powers and Type I errors; with $d=3$ independent components drawn from a Student-t distribution and sample size $n=500$); the full comparison is provided in Appendix E.5 of the revised manuscript.
>
>   | Method | c=0.1 | c=0.2 | c=0.3 | c=0.4 | c=0.5 | c=0.6 | c=0.7 | c=0.8 | c=0.9 | c=1.0 | Type I |
>   | :----: | :---: | :---: | :---: | :---: | :---: | :---: | :---: | :---: | ----- | ----- | ------ |
>   |  HSIC  | 0.01  | 0.04  | 0.05  |  0.1  | 0.17  | 0.28  | 0.37  | 0.53  | 0.72  | 0.83  | 0.05   |
>   | LiNGIC | 0.08  | 0.06  | 0.12  | 0.23  | 0.39  | 0.53  | 0.69  | 0.86  | 0.94  | 0.95  | 0.05   |
>
> - **(Q2.3)** The reviewer wonders whether the methods can be extended to high-dimensional cases.
>
>   **A:** Thank you for raising this point. We address it from the following three folds:
>
>   1. **The scope of our method:** Our primary motivation is to design an independence test tailored to linear non-Gaussian causal discovery. In this framework, e.g., Direct-LiNGAM, GIN, etc., independence tests are applied between scalar random variables, where each dimension of the observed dataset is treated as a node in the causal graph. Therefore, our study focuses on the scalar case, which directly matches the use cases in linear non-Gaussian causal discovery.
>
>   2. **Empirical Evaluation in high-dimensional causal graphs:** In causal discovery, "high dimensionality" typically refers to graphs with many nodes. To evaluate performance in this setting, we additionally conducted experiments on causal graphs with 20, 50, and 100 nodes. The results (below; see details in Appendix E.3) show that our method remains effective in such large-scale causal discovery tasks.
>
>      | Uniform | F1 score (HSIC) | F1 score (Ours) | SHD (HSIC) | SHD (Ours) |
>      | :-----: | :-------------- | --------------- | ---------- | ---------- |
>      |  d=20   | 0.828           | 0.835           | 6.9        | 6.5        |
>      |  d=50   | 0.550           | 0.613           | 58.9       | 51.1       |
>      |  d=100  | 0.363           | 0.454           | 191.0      | 159.0      |
>
>      | Truncnorm | F1 score (HSIC) | F1 score (Ours) | SHD (HSIC) | SHD (Ours) |
>      | :-------: | :-------------- | --------------- | ---------- | ---------- |
>      |   d=20    | 0.422           | 0.489           | 24.3       | 22.3       |
>      |   d=50    | 0.296           | 0.367           | 119.3      | 101.9      |
>      |   d=100   | 0.283           | 0.358           | 249.7      | 208.6      |
>
>   3. **Theoretical potential for high-dimensional extension:** Our current analysis focuses on scalar variables because Theorems 4.2 and 4.3 rely on scalar moment conditions (i.e., constancy of conditional mean and variance). Extending these theoretical statements to vector-valued variables requires different tools, such as characterizing joint invariance across all linear projections, which go beyond the methodological scope of this work. We therefore leave it for future research.

---

> ### Author Response · Authors · 2025-11-25
> **Response to Reviewer GaAN (Part 4/4)**
>
> - **(Q2.4)** The reviewer suggests providing more simulation experiments similar to the setup in Reference [1] to demonstrate generalizability across different graph structures.
>
>   **A:** Thank you for this constructive suggestion. In light of your comment, we conducted new simulations similar to the settings described in **Section 7.1 of Reference [1]** ([[S+06](https://jmlr.org/papers/v7/shimizu06a.html)]). We evaluated our method against the baseline (DirectLiNGAM with HSIC) under varying graph structures (sparse vs. dense), dimensionalities, and non-Gaussian noise distributions. Results are summarized below and detailed in Appendix E.4 in the revision.
>
>   - **Data Generation:** We simulated both **Sparse (S)** graphs (with the average degree 2) and **Dense (D)** graphs (with probability of having an edge $p=0.7$)). We also **vary the dimension of the dataset** (or the number of nodes) within $\lbrace 5, 10, 15 \rbrace$. We used strictly lower-triangular matrices $\mathbf{B}$ with weights sampled uniformly from $[-1.5, -0.5] \cup [0.5, 1.5]$. The disturbance terms were generated from **non-Gaussian distributions** (e.g., Uniform, Student t, and Truncnorm), followed by a random permutation of the causal order.
>   - **Experiment results:** We report the performance mean over 100 trials on datasets with $n=200$ samples. Our method demonstrates strong generalizability and consistently maintains higher performance with respect to F1 score ($\uparrow$) and SHD ($\downarrow$).
>
>   |     Settings      | **Nodes (d)** | **F1 Score (HSIC)** | **F1 Score (Ours)** | **SHD (HSIC)** | **SHD (Ours)** |
>   | :---------------: | ------------- | ------------------- | ------------------- | -------------- | -------------- |
>   | **Student t (D)** | 5             | 0.906               | **0.909**               | 0.93           | **0.86**           |
>   |                   | 10            | 0.717               | **0.754**               | 13.11          | **11.43**          |
>   |                   | 15            | 0.614               | **0.652**               | 43.3           | **39.21**          |
>   | **Student t (S)** | 5             | 0.913               | **0.935**               | 0.58           | **0.43**           |
>   |                   | 10            | 0.833               | **0.847**               | 2.82           | **2.52**           |
>   |                   | 15            | 0.756               | **0.773**               | 7.36           | **6.53**           |
>   |  **Uniform (D)**  | 5             | 0.905               | **0.927**               | 0.92           | **0.72**           |
>   |                   | 10            | 0.620               | **0.700**               | 17.91          | **14.21**          |
>   |                   | 15            | 0.508               | **0.566**               | 53.95          | **48.21**          |
>   |  **Uniform (S)**  | 5             | 0.945               | **0.968**               | 0.49           | **0.36**           |
>   |                   | 10            | 0.916               | **0.927**               | 1.72           | **1.51**           |
>   |                   | 15            | 0.841               | **0.875**               | 4.52           | **3.49**           |
>   | **Truncnorm (D)** | 5             | 0.474               | **0.572**               | 4.78           | **3.95**           |
>   |                   | 10            | **0.435**               | 0.434               | 25.92          | **25.75**          |
>   |                   | 15            | 0.412               | **0.450**               | 63.31          | **59.84**          |
>   | **Truncnorm (S)** | 5             | 0.506               | **0.613**               | 3.52           | **2.81**           |
>   |                   | 10            | 0.442               | **0.556**               | 10.84          | **8.98**           |
>   |                   | 15            | 0.402               | **0.432**               | 19.18          | **18.93**          |
>
> ---
>
> **(Q3)** The reviewer notes that two relevant works were not discussed.
>
> **A:** We thank the reviewer for pointing out these relevant references [[C21\]](https://arxiv.org/pdf/1410.4179) and  [[H+17\]](https://arxiv.org/pdf/1909.10140). We have now included them in the updated manuscript and revised our discussion accordingly.
>
> ---
>
> We want to thank the reviewer again for all the valuable feedback, and we hope the reviewer's questions are properly addressed. Your further feedback would be appreciated, and we hope for the opportunity to respond to it.

---

> > ### Comment · Reviewer_GaAN · 2025-11-28
> > **Response**
> >
> > The authors have addressed my concerns. I would like to increase the score to 6.

---

### Official Review · Reviewer_EbqJ · 2025-10-31

**Soundness:** 3
**Presentation:** 3
**Contribution:** 3
**Rating:** 6
**Confidence:** 3

**Summary:**

The paper is related to causal discovery and proposes a novel independence test for data with linear dependencies and non-Gaussian noise. The kernel-based testing framework comes with asymptotic guarantees. Several numerical experiments for synthetic as well as real-world data are performed and results are compared to different baseline approaches.

**Strengths:**

S1 The considered problem is interesting and relevant.
S2 The paper is technically sound and overall well-written.
S3 The methods used are suitable.
S4 The evaluation and comparison against baselines is convincing.

**Weaknesses:**

W1 The required computational costs and scalability remain somewhat unclear.
W2 Limitations of the paper’s methods could be better discussed.

**Questions:**

What are the required computational costs?

Does the approach scale in the number of components d and the sample size n?

Which model quantity could become a bottleneck?

Could the assumption of linear mixtures be relaxed to more general ones?

---

> ### Author Response · Authors · 2025-11-25
> **Response to Reviewer EbqJ (Part 1/2)**
>
> We sincerely appreciate the reviewer's constructive comments and helpful feedback. Please see below for our response.
>
> ---
>
> **(Q1)** The reviewer wonders whether the assumption of linear mixtures can be relaxed.
>
> **A:** Thank you for this insightful question. We address it from the following points:
>
> 1. **Motivation:** The linear mixture setting is intentionally chosen and is well-motivated in the causal discovery literature. While independence tests exist for linear Gaussian data, monotonic dependencies, and general non-parametric cases, there is a notable lack of testing methods tailored specifically for linear mixtures of non-Gaussian variables. This setting is a cornerstone assumption in many causal discovery methods, such as LiNGAM and its variants. Our goal is to fill this gap by constructing a test that explicitly leverages this parametric assumption. While general non-parametric tests could tackle non-linear mixtures, they typically sacrifice statistical power for generality. By restricting the scope to linear mixtures, we achieve higher testing power for this widespread model class.
> 2. **Theoretical scope:** Theoretically, the linearity assumption is a prerequisite for our characterization of independence. The proof of Theorem 1 relies on the linearity assumption to connect the constancy of the conditional mean and variance to independence. Relaxing this to general nonlinear mixtures breaks this connection: in nonlinear cases, dependence may not be captured by the first two moments but rather by higher-order conditional information (e.g. $\mathbb{E}[Y^3|X]$). Therefore, using only the constancy of regression and homoscedasticity would be insufficient for characterizing independence.
>
> Extending the linear mixture assumption to a more general one is therefore a non-trivial challenge. While we acknowledge this as an interesting direction for future work, we respectfully emphasize that the linear setting here addresses a distinct and critical gap in the current literature.
>
> ------
>
> **(Q2)** The reviewer asks for the computational costs and scalability.
>
> **A:** Thank you for this insightful question. Overall, the computational cost of LiNGIC is $O(n^2)$, where $n$ represents sample size. This has been mentioned in the updated manuscript. We address the reviewer's specific questions as follows:
>
> - **(Q2.1)** What is the required computational cost?
>
>   **A:** Given $n$ samples, computing the $n\times n$ Gram matrices requires $O(n^2)$ kernel evaluations. As in  [[G+05](https://surl.li/svrdhx)], the centering and evaluation of the V-statistic $\operatorname{Tr}(K_XHL_YH)$ can be implemented using three double sums, each of which is $O(n^2)$. Therefore, one LiNGIC evaluation has overall time complexity $O(n^2)$ (quadratic in the sample size), matching the complexity of standard kernel-based independence tests (e.g., HSIC).
>
> - **(Q2.2)** Does the approach scale in the number of components $d$ and the sample size $n$?
>
>   **A:** The number of non-Gaussian components $d$ appears only in the data-generating process (e.g. $Y=\sum_i^da_i\varepsilon_i$), but LiNGIC never estimates these components nor requires access to $d$. Our test only uses the observed variables $X$ and $Y$ and computes a dependence measure between them. Therefore, **the computational cost of LiNGIC is completely independent of $d$**.
>
>   On the other hand, the sample size $n$ does affect the runtime: as discussed above, one LiNGIC evaluation has quadratic time complexity $O(n^2)$ due to the construction and manipulation of $n \times n$ Gram matrices. We also provide empirical runtime results across varying $n$ (shown below and in Appendix E.2 in the revision), which confirm this quadratic scaling.
>
>   | Sample Size \ Method | HSIC  | LiNGIC | RDC   |
>   | :------------------: | :---- | ------ | :---- |
>   |        n=500         | 0.011 | 0.036  | 2.190 |
>   |        n=1000        | 0.051 | 0.150  | 2.478 |
>   |        n=2000        | 0.205 | 0.499  | 3.121 |
>   |        n=3000        | 0.411 | 1.154  | 3.778 |
>   |        n=4000        | 0.678 | 2.016  | 4.253 |
>   |        n=5000        | 0.995 | 3.126  | 4.773 |
>
> - **(Q2.3)** Which model quantity could become a bottleneck?
>
>   **A:** The dominant computational bottleneck of LiNGIC is the construction and storage of the $n\times n$ Gram matrices, which require $O(n^2)$ time and memory. This is also the standard bottleneck for classical kernel-based independence tests such as HSIC.
>
>   Currently, our main contribution is improved statistical power under the linear non-Gaussian setting, rather than computational acceleration. Designing a faster variant of LiNGIC is a potential direction for future work, such as acceleration of the kernel matrix computation using incomplete Cholesky decomposition and Nyström method [[W+00](https://surl.lu/ufdkpb), [D+05](https://www.jmlr.org/papers/volume6/drineas05a/drineas05a.pdf)].

---

> ### Author Response · Authors · 2025-11-25
> **Response to Reviewer EbqJ (Part 2/2)**
>
> **(Q3)** The reviewer asks for a clearer discussion of the limitations of our method.
>
> **A:** Thank you for raising this. We have added a discussion paragraph on limitations in the revised manuscript in Appendix F.2. As mentioned above, we acknowledge that our theoretical guarantees rely on the assumption of linear mixtures of non-Gaussian variables. Extending the setting to more general nonlinear mixing models would be a non-trivial task and represents a promising avenue for future research.
>
> ---
>
> We want to thank the reviewer again for all the valuable feedback.

---

### Official Review · Reviewer_cGuL · 2025-11-01

**Soundness:** 3
**Presentation:** 3
**Contribution:** 2
**Rating:** 4
**Confidence:** 4

**Summary:**

The paper targets independence testing tailored to linear non-Gaussian settings. Its main theoretical observation is a characterization: for linear mixtures of independent non-Gaussian components, $X \mathrel{\unicode{x2AEB}} Y$ holds iff both the conditional mean and the conditional variance are constant. Building on this, the authors design a test statistic (LiNGIC) that couples a universal kernel on $X$ with a quadratic feature/polynomial kernel on $Y$, making it sensitive precisely to departures in those two conditional moments. They derive an asymptotic null and use a Gamma approximation for fast calibration. Empirically, the paper (i) reports Type-I control and power under several non-Gaussian families and increasing mixture complexity, and (ii) demonstrates downstream causal discovery by swapping this test into Direct-LiNGAM. Across synthetic settings, the method shows higher power than standard unconditional tests in the target regime, while keeping computation low thanks to the closed-form calibration.

**Strengths:**

1. The paper targets the dependence between linear non-Gaussian mixtures and proves a specific and clean equivalence between independence and constancy of the first two conditional moments.
2. The theoretical work of the test statistic seems to be solid and complete, although standard results have been given in other papers.
3. The empirical results on pairwise dependence clearly show the advantage of the proposed method, as it generally has higher powers across all regimes.

**Weaknesses:**

1. The proposed method can only be applied to pairwise independence testing. A joint independence test (e.g. dHSIC) between a number of variables would have more potential applications, especially in causal discovery. I wonder whether such an extension could be possible.
2. The computational advantage of the proposed method would be strengthened if the author could provide runtime comparisons of the method on pairwise independence testing and application to causal discovery.
3. In the experiments, the proposed method and HSIC both used the Gamma approximation. Although it alleviates the computational burden, its performance might be reduced. A usual alternative is to use the permutation test, and the author did not discuss this part. Therefore,  results for HSIC and LiNGIC with permutation calibration should be included to see whether LiNGIC + Gamma approximation remains superior.
4. In general, the power issue is more severe in high-dimensional causal discovery problems, i.e., many (joint) independence tests may simply reject dependence if multiple variables are involved. Therefore, I would like to see more experiments on LiNGAM with $d = 20, 50, 100$ with SHD, F1 score comparison to evaluate whether the proposed method brings a significant advantage to this scenario.

**Questions:**

See weaknesses.

---

> ### Author Response · Authors · 2025-11-25
> **Response to Reviewer cGuL (Part 1/2)**
>
> We sincerely appreciate the reviewer's constructive comments and helpful feedback. Please see below for our response.
>
> ---
>
> **(Q1)** The reviewer wonders whether LiNGIC can be extended to a joint independence test.
>
> **A:** Thank you for raising this point. The **short answer is yes**; in our **linear non-Gaussian setting**, joint independence can indeed be reduced to pairwise independence. As stated in [[C+07\]](https://link.springer.com/content/pdf/10.1007/978-3-540-74494-8_2.pdf), Proposition 1:
>
> >  *Let $s$ be a random vector with mutually independent components, and* $x=Gs$. *Then the mutual independence of the entries of* $x$ *is equivalent to their pairwise independence.*
>
> In our framework, the linear mixtures are of the form  $Y=\sum_ia_i\varepsilon_i=a^T\varepsilon$, $X=\sum_ib_i\varepsilon_i=b^T\varepsilon$, and $Z=\sum_ic_i\varepsilon_i=c^T\varepsilon$. So that $(Y,X,Z)^T=[a,b,c]^T\varepsilon\triangleq G\varepsilon$, where $\varepsilon$ has mutually independent (non-Gaussian) components. Hence, the mutual (joint) independence of $(Y,X,Z)$ is equivalent to their pairwise independence under our model.
>
> Therefore, to test the joint independence of $(Y,X,Z)$, one can apply LiNGIC to the pairs $(Y,X), (Y,Z),(X,Z)$ and combine the results accordingly. Extending LiNGIC to a direct multivariate test in this setting (in the spirit of dHSIC) is an interesting direction for future work.
>
> ---
>
> **(Q2)** The reviewer suggests several additional experiments.
>
> - **(Q2.1)** LiNGIC/HSIC results under permutation testing.
>
>   **A:** Thank you for raising this. We have added permutation-calibrated results for both LiNGIC and HSIC, and summarize the results below. These experiments show that the permutation-based method yields almost the same test power and Type I error as the Gamma approximation, and both versions of LiNGIC consistently achieve the highest power among all baselines. Since permutation-based procedures have substantially higher computational cost (see our reply to Q2.2) while providing no noticeable advantage over the Gamma approximation, we use the latter for efficiency.
>
>   |     Method      |    HSIC    |           | HSIC_perm  |           |   LiNGIC   |           | LiNGIC_perm |           |
>   | :-------------: | :--------: | :-------: | :--------: | :-------: | :--------: | :-------: | :---------: | :-------: |
>   |   #components   | **Type I** | **Power** | **Type I** | **Power** | **Type I** | **Power** | **Type I**  | **Power** |
>   |  Laplace (d=2)  |    0.04    |   0.84    |    0.04    |   0.79    |    0.07    |   0.84    |    0.04     |   0.84    |
>   |      (d=3)      |    0.01    |   0.77    |    0.05    |   0.75    |    0.04    |    0.8    |    0.05     |    0.8    |
>   |      (d=4)      |    0.06    |   0.61    |    0.04    |   0.61    |    0.05    |   0.67    |    0.04     |   0.68    |
>   |      (d=5)      |    0.04    |   0.56    |    0.02    |   0.52    |    0.07    |    0.7    |    0.07     |   0.71    |
>   |      (d=6)      |    0.09    |   0.49    |    0.03    |   0.48    |    0.06    |   0.66    |    0.06     |   0.67    |
>   | Student t (d=2) |    0.07    |   0.79    |    0.03    |   0.69    |    0.08    |   0.85    |    0.05     |   0.86    |
>   |      (d=3)      |    0.05    |   0.85    |    0.06    |    0.8    |    0.06    |   0.97    |    0.05     |   0.95    |
>   |      (d=4)      |    0.05    |   0.83    |    0.07    |   0.83    |    0.06    |   0.95    |    0.06     |   0.92    |
>   |      (d=5)      |    0.07    |   0.75    |    0.05    |   0.78    |    0.04    |   0.93    |    0.04     |   0.93    |
>   |      (d=6)      |    0.04    |    0.7    |    0.07    |   0.73    |    0.03    |   0.96    |    0.06     |   0.96    |
>   |  Uniform (d=2)  |    0.07    |   0.87    |    0.07    |   0.82    |    0.07    |   0.88    |    0.07     |   0.88    |
>   |      (d=3)      |    0.04    |   0.76    |    0.07    |   0.72    |    0.05    |   0.83    |    0.05     |   0.82    |
>   |      (d=4)      |    0.02    |   0.47    |    0.03    |    0.5    |    0.06    |   0.54    |    0.05     |   0.53    |
>   |      (d=5)      |    0.05    |   0.37    |    0.02    |   0.35    |    0.06    |   0.45    |    0.06     |   0.45    |
>   |      (d=6)      |    0.03    |   0.24    |    0.04    |   0.14    |    0.03    |   0.31    |    0.05     |   0.29    |
>   | Truncnorm (d=2) |    0.06    |   0.19    |    0.09    |   0.15    |    0.08    |   0.41    |    0.06     |   0.38    |
>   |      (d=3)      |    0.02    |   0.11    |    0.06    |   0.11    |    0.02    |   0.25    |    0.03     |   0.23    |
>   |      (d=4)      |    0.04    |   0.05    |    0.04    |   0.06    |    0.05    |   0.12    |    0.06     |    0.1    |
>   |      (d=5)      |    0.04    |   0.03    |    0.01    |   0.04    |    0.05    |   0.04    |    0.06     |   0.04    |
>   |      (d=6)      |    0.06    |   0.07    |    0.02    |   0.05    |    0.06    |   0.03    |    0.06     |   0.03    |

---

> ### Author Response · Authors · 2025-11-25
> **Response to Reviewer cGuL (Part 2/2)**
>
> - **(Q2.2)** Runtime comparison for pairwise tests and causal discovery.
>
>   **A:** Thank you for this insightful question. Overall, the computational cost of LiNGIC is $O(n^2)$, where $n$ represents sample size. Specifically, given $n$ samples, computing the $n\times n$ Gram matrices requires $O(n^2)$ kernel evaluations. As in  [[G+05\]](https://surl.li/svrdhx), the centering and evaluation of the V-statistic $\operatorname{Tr}(K_XHL_YH)$ can be implemented using three double sums, each of which is $O(n^2)$. Therefore, one LiNGIC evaluation has overall time complexity $O(n^2)$ (quadratic in the sample size), matching the complexity of standard kernel-based independence tests (e.g., HSIC). This has been mentioned in Appendix F.3 in the updated manuscript.
>
>   Importantly, our main contribution is improved statistical power under the linear non-Gaussian setting, rather than computational acceleration. Nevertheless, LiNGIC operates on a comparable runtime scale to the widely used HSIC. Designing a faster variant of LiNGIC is a potential direction for future work.
>
>   For empirical evaluation, we have also compared the runtime (in seconds) of our testing method with others, and their applications in causal discovery methods (here we use Direct-LiNGAM) correspondingly. Results are shown as follows:
>
>   | Sample Size \ Method | HSIC  | HSIC_perm | LiNGIC | LiNGIC_perm | RDC   |
>   | :------------------: | :---- | --------- | ------ | ----------- | ----- |
>   |        n=500         | 0.011 | 2.693     | 0.036  | 7.159       | 2.190 |
>   |        n=1000        | 0.051 | 16.267    | 0.150  | 36.427      | 2.478 |
>   |        n=2000        | 0.205 | 66.178    | 0.499  | 102.564     | 3.131 |
>   |        n=3000        | 0.411 | 166.481   | 1.154  | 233.445     | 3.778 |
>   |        n=4000        | 0.678 | 301.443   | 2.016  | 405.439     | 4.352 |
>   |        n=5000        | 0.995 | 483.046   | 3.126  | 638.219     | 4.773 |
>
>   | Runtime comparison (s) | Direct-LiNGAM (HSIC) | Direct-LiNGAM (LiNGIC) |
>   | :--------------------: | :------------------- | :--------------------- |
>   |      n=200, d=20       | 4.75                 | 7.05                   |
>   |      n=200, d=50       | 62.46                | 99.52                  |
>   |      n=200, d=100      | 500.17               | 802.77                 |
>
>   We observe that LiNGIC operates on a similar timescale as classical kernel-based tests.
>
> - **(Q2.3)** High-dimensional LiNGAM (number of nodes $d=\lbrace 20,50,100 \rbrace$) evaluating SHD/F1.
>
>   **A:** Thank you for this insightful comment. Following your suggestion, we conducted additional simulations to evaluate our method on high-dimensional causal graphs with the number of nodes $d\in\lbrace 20,50,100 \rbrace$. Results are summarized below and detailed in Appendix E.5 in the revision.
>
>   - **Data Generation:** We simulated graphs with an average degree of 2. We used strictly lower-triangular matrices $\mathbf{B}$ with weights sampled uniformly from $[-1.5, -0.5] \cup [0.5, 1.5]$. The disturbance terms were generated from Uniform or Truncnorm, followed by a random permutation of the causal order.
>
>   - **Experiment results:** We report the performance mean over 100 trials on datasets with $n=200$ samples. Our method demonstrates strong generalizability and consistently maintains better performance with respect to F1 score ($\uparrow$) and SHD ($\downarrow$).
>
>     | Uniform | F1 score (HSIC) | F1 score (Ours) | SHD (HSIC) | SHD (Ours) |
>     | :-----: | :-------------- | --------------- | ---------- | ---------- |
>     |  d=20   | 0.828           | **0.835**           | 6.9        | **6.5**        |
>     |  d=50   | 0.550           | **0.613**           | 58.9       | **51.1**       |
>     |  d=100  | 0.363           | **0.454**           | 191.0      | **159.0**      |
>
>     | Truncnorm | F1 score (HSIC) | F1 score (Ours) | SHD (HSIC) | SHD (Ours) |
>     | :-------: | :-------------- | --------------- | ---------- | ---------- |
>     |   d=20    | 0.422           | **0.489**           | 24.3       | **22.3**       |
>     |   d=50    | 0.296           | **0.367**           | 119.3      | **101.9**      |
>     |   d=100   | 0.283           | **0.358**           | 249.7      | **208.6**      |
>
> ---
>
> Once again, we are grateful for the reviewer's valuable comments, and sincerely hope that you will find your suggestions and concerns well addressed by the response above and the relevant sections in the paper. Your further feedback would be appreciated, and we hope for the opportunity to respond to it.

---

> > ### Comment · Reviewer_cGuL · 2025-11-27
> >
> > I thank the author for the detailed rebuttal which has addressed all my concerns. I am happy to increase my score to 6.

---

### Official Review · Reviewer_FrNB · 2025-11-11

**Soundness:** 3
**Presentation:** 2
**Contribution:** 2
**Rating:** 4
**Confidence:** 2

**Summary:**

This paper proposes a new independence test, LiNGIC, specifically for linear non-Gaussian (LiNGAM) data. It argues that general tests like HSIC lack statistical power in this structured setting. The core contribution is Theorem 4.2, which proves that for LiNGAM data, independence is equivalent to the conditional mean and conditional variance both being constant. The LiNGIC statistic is a kernel-based test designed to check these two conditions simultaneously. Experiments demonstrate that LiNGIC achieves higher statistical power than baselines and improves downstream causal discovery performance.

**Strengths:**

- The paper provide strong theoretical results.
- In general, the paper is well-written.

**Weaknesses:**

- Although the authors have made efforts to justify developing conditional independence tests under the strong linear non-Gaussian assumption, I remain unconvinced. In practice, it is difficult to verify that real-world data strictly satisfies this assumption. Moreover, it is not entirely fair to compare their method with approaches that assume general nonparametric settings, as the learning problem is substantially simpler under the linear non-Gaussian constraint.

- The evaluation of causal discovery performance is limited, with insufficient analysis across different graph structures, noise types, and data settings. Key dataset details (e.g., observational vs. interventional) are also unclear.

- The paper does not include comparisons with general nonparametric conditional independence tests or established causal discovery methods, limiting the empirical context of the results.

**Questions:**

1. The authors suggest that in certain applications, real-world data follows a linear non-Gaussian setting. Can they provide a concrete example? In particular, the proposed method is evaluated on the Sachs dataset, yet there is no evidence supporting a linear non-Gaussian assumption. In fact, methods designed for nonlinear settings often achieve better performance, indicating that the Sachs dataset may not be an ideal benchmark for this paper.

2. While the paper emphasizes applications in causal discovery, the evaluation in this context is quite limited. I recommend providing detailed causal discovery results under various graph structures and data settings. Additionally, the description of the Sachs dataset in the appendix is insufficient. Which type of data was evaluated: observational, interventional, or both?

3. To make the evaluation more comprehensive, the authors should consider including results from general nonparametric constraint-based conditional independence tests or causal discovery methods, such as:

> [1] Strobl, Eric V., Kun Zhang, and Shyam Visweswaran. "Approximate kernel-based conditional independence tests for fast non-parametric causal discovery." Journal of Causal Inference 7.1 (2019): 20180017.

> [2] Zarebavani, Behrooz, et al. "cuPC: CUDA-based parallel PC algorithm for causal structure learning on GPU." IEEE Transactions on Parallel and Distributed Systems 31.3 (2019): 530-542.

> [3] Zhang, Hao, et al. "Residual similarity based conditional independence test and its application in causal discovery." Proceedings of the AAAI conference on artificial intelligence. Vol. 36. No. 5. 2022.

4. The "Application in Causal Discovery" (Section 6.2) shows mixed results. Although the proposed method performs best on data with "Uniform" noise, it fails on "Laplace," "Student t," and "TruncNorm" noise, achieving F1 scores near 0.0—comparable to other methods. This is a significant weakness, as it contradicts Figures 3 and 4, which show that LiNGIC has superior statistical power for these distributions. The paper does not address this discrepancy.

---

> ### Author Response · Authors · 2025-11-25
> **Response to Reviewer FrNB (Part 1/3)**
>
> We appreciate the reviewer's positive assessment and the detailed feedback. We noticed that some concerns seem to stem from a perspective of **conditional independence testing**. Since our proposed method is designed for **unconditional independence**, we clarify this distinction in our point-by-point responses below.
>
> ---
>
> **(Q1)** The reviewer has concerns regarding the linear non-Gaussian assumption.
>
> - **(Q1.1)** Validity of the Linear Non-Gaussian Assumption on Real Data (e.g. Sachs Dataset).
>
>   **A:** Thank you for this insightful question. We agree that real-world data rarely allows us to definitely verify whether the linear non-Gaussian assumption holds, since we can not assume causal sufficiency and no selection bias. This is a common challenge in causal discovery: even if the underlying data-generating process were linear and non-Gaussian, preprocessing steps, measurement noise, and selection effects can easily induce nonlinearities or obscure non-Gaussianity.
>
>   However, **there exist many successful applications of methods with linear non-Gaussian assumptions** achieving great results in real-world tasks. For instance, in neuroscience, EEG/MEG source separation is routinely modeled using linear mixing with non-Gaussian independent components (called ICA)  [[M+95](https://surl.li/aohufo), [H00](https://www.sciencedirect.com/science/article/abs/pii/S0893608000000265), [D+07](https://www.sciencedirect.com/science/article/abs/pii/S1053811906011098)]. This is the most widely used real-world linear non-Gaussian application. We also added a section in the appendix to discuss it further (Appendix F.3).
>
>   Regarding the Sachs dataset, it has historically been used as a benchmark for linear and nonlinear causal discovery methods. Although the biochemical interactions in this system may not be linear, several prior works have shown that linear non-Gaussian methods can still recover meaningful causal relations [[D+22](https://arxiv.org/abs/2210.11021)]. These results recovered from them offer a partial empirical validation for model adequacy.
>
> - **(Q1.2)**  It is not fair to compare with approaches assuming general nonparametric settings.
>
>   **A:** Thank you for raising this. While the linear non-Gaussian setting is indeed more restrictive than a fully nonparametric model, we consider our comparisons remain appropriate and meaningful for the following reasons.
>
>   The linear non-Gaussian setting is a well-established assumption rather than an arbitrary simplification. Empirically, linear models often serve as robust approximations in many fields, e.g., economics [[I+94](https://www.jstor.org/stable/2951620), [W+10](https://www.jstor.org/stable/j.ctt5hhcfr)]. Additionally, in biology [[H14](https://www.scirp.org/reference/referencespapers?referenceid=2050944)], simple transformations such as log scales can make biological relationships nearly linear. The non-Gaussian assumption is also mild and broadly applicable. Real-world data are frequently non-Gaussian, and non-Gaussianity has long been recognized as a powerful signal exploited in ICA and blind source separation [[H99](https://www.cs.helsinki.fi/u/ahyvarin/papers/TNN99new.pdf), [H00](https://www.sciencedirect.com/science/article/abs/pii/S0893608000000265)]. In many causal discovery applications, especially in physics and social sciences, linear non-Gaussian models have proven to be effective and highly practical [[S+06](https://www.jmlr.org/papers/volume7/shimizu06a/shimizu06a.pdf)].
>
>   Importantly, if the data are (approximately) linear non-Gaussian, then **leveraging this information is not only natural but also beneficial**. Using a fully nonparametric test in such cases ignores available structure and leads to unnecessarily complex and less powerful testing procedures. This follows the classical principle that one should not solve a more general problem than necessary [[V00](https://link.springer.com/book/10.1007/978-1-4757-3264-1)].
>
>   Therefore, given the prevalence of such linear non-Gaussian data, it is important to develop an independence test tailored specifically to the linear non-Gaussian setting. To the best of our knowledge, **no prior work has addressed this problem, and ours is the first independence test explicitly designed for this parametric setting**. For this reason, it is natural and appropriate to use existing nonparametric independence tests as baselines in our evaluation.

---

> > ### Author Response · Authors · 2025-11-25
> > **Response to Reviewer FrNB (Part 2/3)**
> >
> > **(Q2)** The reviewer asks for clarification on several experimental details and suggests several additional experiments.
> >
> > - **(Q2.1)** The description of the SACHS dataset is insufficient.
> >
> >   **A:** Thank you for pointing this out. We apologize for the confusion. To clarify, the SACHS dataset consists of $n=853$ **observational** measurements of 11 proteins. A more detailed description has been added to **Appendix E.3**, and we explicitly reference it in the main text for clarity.
> >
> > - **(Q2.2)** Include more **conditional independence tests** and causal discovery methods.
> >
> >   **A:** We thank the reviewer for the suggestion to broaden our comparisons. As our method is formulated for **unconditional independence**, standard **conditional independence tests** are not directly applicable as baselines. Therefore, we have included the **corresponding unconditional version of those conditional independence tests**.
> >
> >   1. **RCIT / RCoT** [[1](https://surl.li/imkelv)]: These are accelerations of KCIT using random Fourier features (RFF) to approximate kernel matrices. In the **unconditional setting, RCIT reduces to HSIC-RFF** [[Z+18](https://surli.cc/epfpgh)], which is already included in our experiments.
> >
> >   2. **cuPC** [[2](https://surl.li/ovwpeo)]: It is a GPU-accelerate implementation of the PC-stable algorithm and relies on **conditional independence testing**. Since our contribution is a **new unconditional independence test**, comparisons with structural learning pipelines such as cuPC fall outside the scope of this work.
> >
> >   3. **SCIT** [[3](https://surl.li/frblfb)]: SCIT is designed for **conditional independence testing** under linear non-Gaussian SEMs by regressing out conditioning variables and applying a **generic nonparametric kernel-based independence check** on residuals. When setting $Z=\emptyset$, SCIT reduces to testing whether the coordinate-wise RBF similarity between $(x,y)$ is significantly larger than that between $(x,r)$, where $r$ is a permuted copy of $y$. This generic independence test does not leverage the linear non-Gaussian structure as our method does. We added SCIT with $Z=\emptyset$  as one of our baselines in the revision and list part of the results here.
> >
> >      |     Method      |    HSIC    |           |    SCIT    |           |   LiNGIC   |           |
> >      | :-------------: | :--------: | :-------: | :--------: | :-------: | :--------: | :-------: |
> >      |   #components   | **Type I** | **Power** | **Type I** | **Power** | **Type I** | **Power** |
> >      |  Laplace (d=2)  |    0.04    |   0.84    |    0.07    |   0.57    |    0.07    |   0.84    |
> >      |      (d=3)      |    0.01    |   0.77    |    0.01    |   0.61    |    0.04    |    0.8    |
> >      |      (d=4)      |    0.06    |   0.61    |    0.01    |    0.6    |    0.05    |   0.67    |
> >      |      (d=5)      |    0.04    |   0.56    |    0.06    |   0.51    |    0.07    |    0.7    |
> >      |      (d=6)      |    0.09    |   0.49    |    0.05    |   0.39    |    0.06    |   0.66    |
> >      | Student t (d=2) |    0.07    |   0.79    |    0.02    |   0.54    |    0.08    |   0.85    |
> >      |      (d=3)      |    0.05    |   0.85    |    0.06    |   0.69    |    0.06    |   0.97    |
> >      |      (d=4)      |    0.05    |   0.83    |    0.03    |   0.62    |    0.06    |   0.95    |
> >      |      (d=5)      |    0.07    |   0.75    |    0.06    |    0.6    |    0.04    |   0.93    |
> >      |      (d=6)      |    0.04    |    0.7    |    0.07    |   0.57    |    0.03    |   0.96    |
> >      |  Uniform (d=2)  |    0.07    |   0.87    |    0.05    |     0     |    0.07    |   0.88    |
> >      |      (d=3)      |    0.04    |   0.76    |    0.07    |   0.01    |    0.05    |   0.83    |
> >      |      (d=4)      |    0.02    |   0.47    |    0.06    |     0     |    0.06    |   0.54    |
> >      |      (d=5)      |    0.05    |   0.37    |    0.03    |     0     |    0.06    |   0.45    |
> >      |      (d=6)      |    0.03    |   0.24    |    0.04    |     0     |    0.03    |   0.31    |
> >      | Truncnorm (d=2) |    0.06    |   0.19    |    0.12    |   0.01    |    0.08    |   0.41    |
> >      |      (d=3)      |    0.02    |   0.11    |    0.01    |   0.01    |    0.02    |   0.25    |
> >      |      (d=4)      |    0.04    |   0.05    |    0.06    |   0.01    |    0.05    |   0.12    |
> >      |      (d=5)      |    0.04    |   0.03    |    0.04    |     0     |    0.05    |   0.04    |
> >      |      (d=6)      |    0.06    |   0.07    |    0.05    |     0     |    0.06    |   0.03    |

---

> ### Author Response · Authors · 2025-11-25
> **Response to Reviewer FrNB (Part 3/3)**
>
> - **(Q2.3)** More causal discovery experiments on different settings.
>
>   **A:** Thank you for this constructive suggestion. We conducted new simulations similar to the settings described in Section 7.1 of ([[S+06](https://jmlr.org/papers/v7/shimizu06a.html)]). We evaluated our method against the baseline (DirectLiNGAM with HSIC) under varying graph structures (sparse vs. dense), dimensionalities, and non-Gaussian noise distributions. Results are summarized below and detailed in Appendix E.4 in the revision.
>
>   - **Data Generation:** We simulated both **Sparse (S)** graphs (with the average degree 2) and **Dense (D)** graphs (with a probability of having an edge $p=0.7$)). We also vary the dimension of the dataset (or the number of nodes) within $\lbrace 5,10,15 \rbrace$. We used strictly lower-triangular matrices $\mathbf{B}$ with weights sampled uniformly from $[-1.5, -0.5] \cup [0.5, 1.5]$. The disturbance terms were generated from **non-Gaussian distributions** (e.g., Uniform, Student t, and Truncnorm), followed by a random permutation of the causal order.
>   - **Experiment results:** We report the performance mean over 100 trials on datasets with $N=200$ samples. Our method demonstrates strong generalizability and consistently maintains higher performance with respect to the F1 score ($\uparrow$) and SHD ($\downarrow$).
>
>   |     Settings      | **Nodes (d)** | **F1 Score (HSIC)** | **F1 Score (Ours)** | **SHD (HSIC)** | **SHD (Ours)** |
>   | :---------------: | ------------- | ------------------- | ------------------- | -------------- | -------------- |
>   | **Student t (D)** | 5             | 0.906               | 0.909               | 0.93           | 0.86           |
>   |                   | 10            | 0.717               | 0.754               | 13.11          | 11.43          |
>   |                   | 15            | 0.614               | 0.652               | 43.3           | 39.21          |
>   | **Student t (S)** | 5             | 0.913               | 0.935               | 0.58           | 0.43           |
>   |                   | 10            | 0.833               | 0.847               | 2.82           | 2.52           |
>   |                   | 15            | 0.756               | 0.773               | 7.36           | 6.53           |
>   |  **Uniform (D)**  | 5             | 0.905               | 0.927               | 0.92           | 0.72           |
>   |                   | 10            | 0.620               | 0.700               | 17.91          | 14.21          |
>   |                   | 15            | 0.508               | 0.566               | 53.95          | 48.21          |
>   |  **Uniform (S)**  | 5             | 0.945               | 0.968               | 0.49           | 0.36           |
>   |                   | 10            | 0.916               | 0.927               | 1.72           | 1.51           |
>   |                   | 15            | 0.841               | 0.875               | 4.52           | 3.49           |
>   | **Truncnorm (D)** | 5             | 0.474               | 0.572               | 4.78           | 3.95           |
>   |                   | 10            | 0.435               | 0.434               | 25.92          | 25.75          |
>   |                   | 15            | 0.412               | 0.450               | 63.31          | 59.84          |
>   | **Truncnorm (S)** | 5             | 0.506               | 0.613               | 3.52           | 2.81           |
>   |                   | 10            | 0.442               | 0.556               | 10.84          | 8.98           |
>   |                   | 15            | 0.402               | 0.432               | 19.18          | 18.93          |
>
> - **(Q2.4)** Unexpected discrepancy between discovery results and statistical power.
>
>   **A:** Thank you for pointing out this discrepancy. Upon rechecking our experimental pipeline, we identified an issue in the results recording step for Section 6.2, which caused some of the reported values to be incorrect. We have corrected this issue, re-verified all causal discovery experiments, and updated Tables 1, 3, and 4. The corrected results are consistent with the statistical-power findings in Figure 3-4 and do not affect the conclusions of the paper.
>
> ---
>
> Once again, we thank the reviewer for the insightful feedback, and hope the questions have been properly addressed. Your further feedback would be appreciated, and we hope for the opportunity to respond to it.

---

### Author Response · Authors · 2025-12-02
**Author Summary Comment**

Dear Area Chair and Reviewers,

We truly appreciate your extra time and effort during the unexpected situation. While the discussion period is unfortunately not possible, we hope the following summary offers a clear and fair account of our key responses and the earlier exchange, and may assist in the Area Chair's assessment.

------

We first summarize our main contributions. Our submission provides a focused and well-motivated framework for independence testing under the linear non-Gaussian mixture model, a foundational setting in causal discovery:

1. A new independence characterization showing that constancy of conditional mean and variance fully determines independence (Theorem 4.2).
2. A principled test statistic with exact asymptotic distributions and formal equivalence to this characterization (Sections 4.2–4.3; Theorem 4.3).
3. Extensive synthetic and real-data experiments demonstrating consistently higher power than existing approaches.

We appreciate the reviewers’ positive recognition of these contributions, for example:

- "represents a clear conceptual advancement,"
- "theoretical derivations in the paper are rigorous and logically coherent,"
- "empirical results...clearly show the advantage of the proposed method,"
- "the evaluation and comparison against baselines is convincing," and
- "providing a strong complement to existing independence testing methods."

---

**Common Themes Addressed Across Reviewers**

- **Linear Non-Gaussian Assumption (Reviewer FrNB & EbqJ):**  We explain the motivation and empirical validity of this assumption and provide discussions on real-world applications (Appendix F.3).
- **More Causal Discovery Experiments (Reviewer FrNB & GaAN):** We added new experiments across varying graph structures, dimensions, and noise types, showing improved F1/SHD over HSIC.
- **Performance Using Permutation instead of Gamma Approximation (Reviewer cGuL & GaAN):** We added permutation-calibrated results for LiNGIC and HSIC, which match the Gamma approximation while being significantly slower.
-  **Computational Cost and Runtime Analysis (Reviewer cGuL & EbqJ):** We provided theoretical and empirical runtime analyses confirming our method's $O(n^2)$ complexity comparable to the standard kernel-based test method.
- **High-Dimensional Experiments (Reviewer cGuL & GaAN):** We added high-dimensional causal discovery experiments (up to $d=100$), where LiNGIC yields consistent gains in F1 and SHD.

---

**Reviewer-Specific Points**

**Reviewer FrNB:**

We respectfully note that the reviewer's W1 & Q3 appear to stem from interpreting our method through the lens of **conditional independence testing**, whereas our contribution is a **new unconditional independence test**. We clarified this distinction in our response and expanded experiments to avoid any misunderstanding.

- **Additional CI & CI-based Causal Discovery Baselines:** Since conditional-independence tests are not directly comparable in our setting, we added their appropriate unconditional variants, where our method consistently achieves higher power.

------

**Reviewer  cGuL:**

- **Extension to Joint Independence Test:** We clarified that in linear non-Gaussian mixtures, joint independence reduces to pairwise independence, so LiNGIC has the potential to be extended accordingly.

We also appreciate that Reviewer cGuL mentioned our rebuttal "addressed all my concerns" and "happy to increase my score to 6" after reading our rebuttal and discussion.

---

**Reviewer  EbqJ:**

- **Computational Bottleneck:** We discussed the main computational bottleneck (the $n\times n$ Gram matrices) and pointed to potential accelerations such as Nyström and incomplete Cholesky decomposition methods.

---

**Reviewer GaAN:**

- **Clarification and Reference Suggestion:** We clarified the experimental setup (metrics, ground-truth graph, kernel choices), streamlined the presentation, and incorporated the missing related works.
- **Additional UI Baselines:** We added RDC [[L+13\]](https://arxiv.org/pdf/1304.7717) and MI [[B+19\]](https://arxiv.org/pdf/1711.06642) as new baselines, with results showing LiNGIC achieves consistently higher power across settings.
- **Simulation with Varying Dependence Strength:** We ran new experiments varying dependence strength, confirming LiNGIC’s superior sensitivity across a wide range of effect sizes.

We also appreciate that Reviewer GaAN mentioned our rebuttal "addressed my concerns" and "would like to increase my score to 6" after reading our rebuttal and discussion.

---

There are also many other insightful comments, which we address in detail in our full responses.

We thank the Area Chair and all reviewers again for all the efforts and valuable feedback.

---

### Meta-Review · Area_Chair_eJ8r · 2025-12-06

**Summary:**

This paper proposes LiNGIC, a new independence test tailored to linear non-Gaussian data, a setting central to causal discovery. The authors establish a key theoretical result: in linear mixtures of independent non-Gaussian components, independence holds if and only if both conditional mean and conditional variance are constant. Leveraging this, LiNGIC combines a universal kernel on inputs with a quadratic kernel on outputs to detect deviations in these two moments. The test has a tractable asymptotic null distribution, enabling fast calibration. Experiments on synthetic and real datasets demonstrate strong Type-I error control, higher power than competing tests, and improved performance when integrated into Direct-LiNGAM.

Causal discovery in linear non-Gaussian settings is a well-studied topic, and this paper introduces a new nonparametric approach with solid theoretical guarantees. Reviewers raised concerns regarding computational efficiency and high-dimensional scenarios, which the authors addressed satisfactorily during the rebuttal. Since most reviewers acknowledge the contribution of this work and support acceptance, I would also recommend accepting the paper.

**Reviewer Concerns:**

Linear Non-Gaussian Assumption (Reviewer FrNB & EbqJ), More Causal Discovery Experiments (Reviewer FrNB & GaAN):
Performance Using Permutation instead of Gamma Approximation (Reviewer cGuL & GaAN),Computational Cost and Runtime Analysis (Reviewer cGuL & EbqJ), High-Dimensional Experiments (Reviewer cGuL & GaAN).

As summarized by the authors the above were the main concerns raised by the reviewers and they are well addressed during the rebuttal.

**Reviewer Scores:**

The authors clearly addressed the concerns and they would increase thier score.

---

### Decision · Program_Chairs · 2026-01-26

Accept (Poster)